# Parallel Stochastic Gradient-Based Planning for World Models

**Michael Psenka** [1 2 ∘]  **Michael Rabbat** [2]  **Aditi Krishnapriyan** [1]  **Yann LeCun** [* 2 3]  **Amir Bar** [* 2]

## Abstract

World models simulate environment dynamics from raw sensory inputs like video. However, using them for planning can be challenging due to the vast and unstructured search space. We propose a robust and highly parallelizable planner that leverages the differentiability of the learned world model for efficient optimization, solving long-horizon control tasks from visual input. Our method treats states as optimization variables ("virtual states") with soft dynamics constraints, enabling parallel computation and easier optimization. To facilitate exploration and avoid local optima, we introduce stochasticity into the states. To mitigate sensitive gradients through high-dimensional vision-based world models, we modify the gradient structure to descend towards valid plans while only requiring action-input gradients. Our approach can be viewed as a stochastic version of a non-condensed or collocation-based optimal controller. We provide theoretical justification and experiments on video-based world models, where our resulting planner outperforms existing planning algorithms like the cross-entropy method (CEM) and vanilla gradient-based optimization (GD) on long-horizon experiments, both in success rate and time to convergence. Source code is available from the project website: https://www.michaelpsenka.io/grasp.

## 1. Introduction

Intelligent agents carry a small-scale model of external reality which allows them to simulate actions, reason about their consequences, and choose the ones that lead to the best outcome. Attempts to build such models date back to control theory, and in recent years researchers have made progress in building world models using deep neural networks trained directly from the raw sensory input (e.g., vision). For example, recent world models have shown success modeling computer games (Valevski et al., 2024), navigating real-world environments (Bar et al., 2025; Ball et al., 2025), and robot arm motion commands (Goswami et al., 2025). World models have numerous impactful applications from simulating complex medical procedures (Koju et al., 2025) to testing robots in visually realistic environments (Guo et al., 2025).

Current planning algorithms used with world models often rely on $0^{\text{th}}$-order optimization methods such as the Cross-Entropy Method (CEM (Rubinstein & Kroese, 2004)) or in general *shooting methods* which plan by iteratively rolling out trajectories and choosing actions from the optimal rollout (Bock & Plitt, 1984; Piovesan & Tanner, 2009). These approaches are simple and robust, but their performance degrades with longer planning horizons and higher action dimensionality (Bharadhwaj et al., 2020), motivating the use of gradient information when differentiable world models are available.

Gradient-based planners exploit the differentiability of learned world models to directly optimize action sequences, enabling more sample-efficient planning and finer-grained improvement than zero-order methods. However, there are two main challenges to this optimization approach: local minima (Jyothir et al., 2023) and instability due to differentiating through the full rollout, akin to backpropagation through time (Werbos, 2002). While prior methods have formulated the optimization for better conditioning (e.g., multiple shooting and direct collocation (Von Stryk, 1993)), these techniques are typically developed for known dynamical systems and do not scale as well to deep neural dynamics (often in latent spaces) with brittle or poorly calibrated input-to-output Jacobians.

In this work, we introduce a novel gradient-based planning method for learned world models that decouples temporal dynamics into parallel optimized states (rather than serial rollouts) while remaining robust at long horizons and in high-dimensional state spaces. Rather than planning exclusively through a deep, sequential rollout of the dynamics model, our approach optimizes over *lifted intermediate states* that are treated as independent optimization variables.

---

[*]Equally advised. [∘]Work done at Meta. [1]University of California, Berkeley [2]Meta FAIR [3]New York University. Correspondence to: Michael Psenka <psenka@eecs.berkeley.edu>.

*Proceedings of the $43^{rd}$ International Conference on Machine Learning*, Seoul, South Korea. PMLR 306, 2026. Copyright 2026 by the author(s).

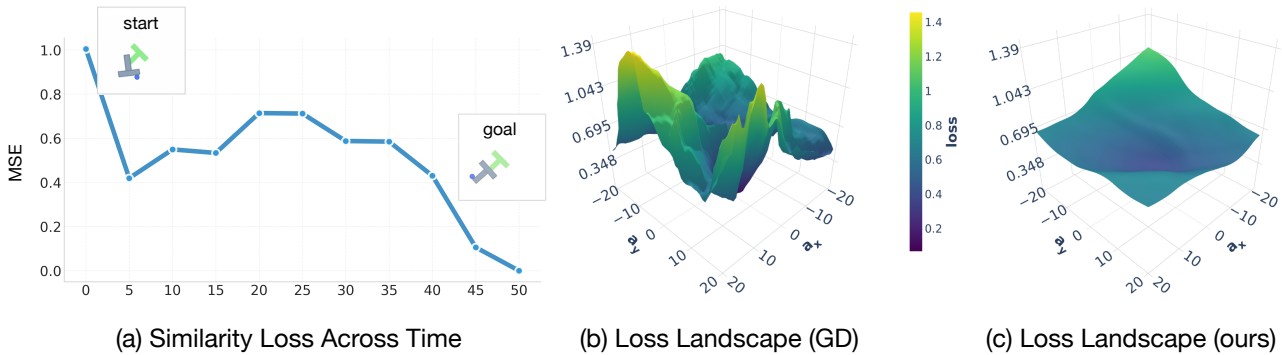

*Figure 1. Difficulty of the planning problem.* Subfigure **(a)** shows the distance to the goal in $L^2$ norm throughout a successful trajectory. This illustrates the difficulty of planning optimization away from a minimizer: successful trajectories often have to first move away from the goal to successfully plan towards it later, resulting in greedy strategies failing. Subfigures **(b)-(c)** depict the loss landscape at convergence of standard rollout-based planners vs. our planner. The example given is in the Push-T environment at horizon length 50. The axes plotted over are with respect to two random, orthogonal, unit-norm directions in the full action space $\mathbb{R}^{50 \times 2}$. Our planner loss is taken as in (10), and for GD the loss is taken as in (3).

Our approach makes two fundamental additions that help solve issues for gradient-based planning in higher dimensions: gradient sensitivity and local minima.

Firstly, a fundamental difficulty arises in the setting of vision-based world models. In high-dimensional learned state spaces, gradients with respect to state inputs can be brittle or adversarial, allowing the optimizer to exploit sensitive Jacobian structure rather than discovering physically meaningful transitions. To mitigate this issue, our planner deliberately stops gradients through the state inputs of the world model, while retaining gradients with respect to actions, which we find behave more reasonably. This alone would promote trajectories near the starting state, but with a dense one-step goal loss over the full trajectory, converged trajectories for these noisy iterations tend towards the goal.

Secondly, to address the remaining non-convexity of the lifted state approach, our planner incorporates Langevin-style stochastic updates on the lifted state variables, explicitly injecting noise during optimization to promote exploration of the state space and facilitate escape from unfavorable basins. This stochastic relaxation allows the planner to search over diverse intermediate trajectories while still favoring solutions that approximately satisfy the learned dynamics. Finally, we intermittently apply a small GD step to fine-tune stochastically optimized trajectories towards fully-optimized paths.

Together, these components yield a practical gradient-based planner for learned visual dynamics that remains stable at long horizons while avoiding the failure modes commonly encountered when backpropagating through deep world-model rollouts. We call our planner **GRASP** (Gradient RelAxed Stochastic Planner) to emphasize its primary components: using gradient information, relaxing the dynamics

constraints, and stochastic optimization for exploration. In various settings, we achieve up to +10% success rate at less than half the compute time cost. We also provide a theoretical model for our planner to further illustrate its role. We demonstrate our planner on visual world models trained on problems in D4RL (Fu et al., 2020) and the DeepMind control suite (Tassa et al., 2018).

## 2. Problem formulation

Our main object of interest is a learned world model $F_\theta : \mathcal{S} \times \mathcal{A} \to \mathcal{S}$ that predicts the next state given the current state and action. For visual domains, states are typically represented in a learned latent space to handle high-dimensional observations. Here we assume $\mathcal{A} = \mathbb{R}^k$ is a continuous Euclidean action space.

We consider the problem of fixed-goal path planning: finding an action sequence $\mathbf{a} = (a_0, a_1, \ldots, a_{T-1})$ that, with respect to the dynamics of the world model $F_\theta$ and a given initial state $s_0 \in \mathcal{S}$, reaches a set goal state $g \in \mathcal{S}$:

$$\mathcal{F}_\theta^T(\mathbf{a}, s_0) = g, \tag{1}$$

where the terminal state $s_T$ is generated recursively through the update rule $s_{t+1} = F_\theta(s_t, a_t)$:

$$\mathcal{F}_\theta^T(\mathbf{a}, s_0) \coloneqq \underbrace{F_\theta(\ldots F_\theta(F_\theta(s_0, a_0), a_1), \ldots a_{T-1})}_{T \text{ times}}. \tag{2}$$

We can sufficiently compute $\mathbf{a}^*$ by solving the following optimization problem:

$$\mathbf{a}^* = \arg\min_{\mathbf{a}} \|\mathcal{F}_\theta^T(\mathbf{a}, s_0) - g\|_2^2. \tag{3}$$

Optimizing (3) directly is challenging due to two main problems. First, it requires $T$ applications of $F_\theta$ (see Eq. 2),

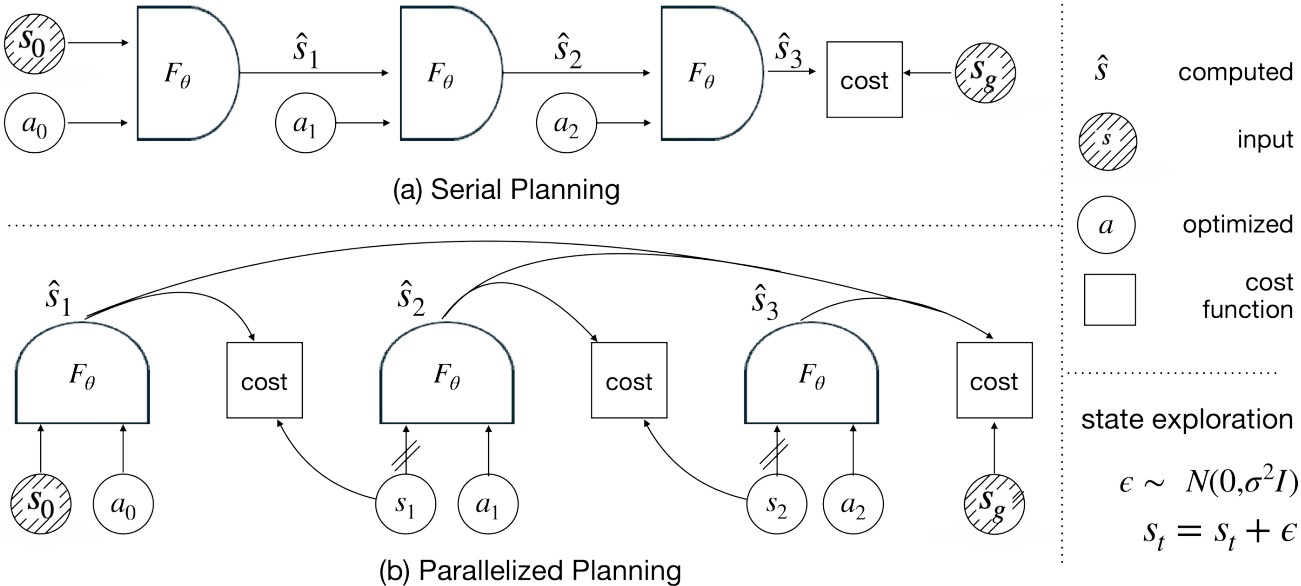

(a) Serial Planning

(b) Parallelized Planning

*Figure 2.* Graphical depiction of **(a)** a standard serial-based setup for optimization-based planning, where states are rolled out using the actions and the loss is evaluated on the goal state, **(b)** our setup, which parallelizes the world model evaluations by optimizing "virtual states" directly and only supervising pairwise dynamics satisfaction. The crossed lines and skipped connections for our method's depiction **(b)** are detailed in Section 3.3, which keeps the full planning graph connected while not requiring state gradients of the dynamics $F_\theta$. For our planner, we find it helpful to alternate between **(a)** and **(b)** throughout the planning optimization.

which is computationally expensive and difficult to optimize due to the poor conditioning arising from repeated applications of $F_\theta$ (see Appendix A.1 for details). Second, it is susceptible to local minima and a jagged loss landscape— see Figure 1. Due to these reasons, existing planners are based on zero-order optimization algorithms like CEM and MPPI (Williams et al., 2016) which are highly stochastic and do not require gradient computation.

In what follows, we propose a gradient-based planner which alleviates these difficulties, while also using the differentiability of the model $F_\theta$. In Section 3, we lift the optimization problem by also optimizing over states, which leads to faster convergence and better conditioning. In Section 3.2, we introduce stochasticity, which helps escape local minima.

## 3. Decoupling dynamics for gradient-based planning

We consider planning with a world model $F_\theta$ and horizon $T$. Given an initial state $s_0 \in \mathcal{S}$ and goal state $g \in \mathcal{S}$, we optimize an action sequence $\mathbf{a} = (a_0, \ldots, a_{T-1})$ such that the rolled-out state $s_T(\mathbf{a})$ is as close to $g$ as possible. A standard approach defines a trajectory by rolling out the model, but backpropagating through a deep composition of $F_\theta$ can be unstable and ill-conditioned. Following prior work on lifted planning (Tamimi & Li, 2009; Rybkin et al., 2021), we introduce auxiliary states $\mathbf{z} = (z_1, \ldots, z_T)$ and enforce dynamics consistency through a penalty function.

### 3.1. Parallelized planning

We first want to decouple the states from the explicit rollout outputs. Writing (3) in terms of each intermediate dynamics condition, we get the following:

$$\min_{\mathbf{s},\mathbf{a}} \ \mathcal{L}_{\text{dyn}}(\mathbf{s}, \mathbf{a}) := \sum_{t=0}^{T-1} \left\| F_\theta(s_t, a_t) - s_{t+1} \right\|_2^2, \quad (4)$$

$$\text{with } s_0 \text{ fixed, } s_T = g.$$

The minimization in (4) is equivalent to (3) in that they share global minimizers.

Immediately, this gives a great benefit in that *all world model evaluations are parallel*; there is no need to do serial rollouts like what is required in (3). There are however two main issues with optimizing this loss directly:

1. *Local minima.* When optimizing with respect to states, the states might be stuck in an unphysical region; for example, Figure 7 shows a case where states go straight through a barrier. To address this, we propose to use Langevin state updates which promote exploration (see Section 3.2).

2. *World model sensitivity for high-dimensional states.* When optimizing $s$ directly over a higher dimensional space (e.g. vision-based), we observe that the Jacobian $J_s F_\theta(s, a)$ does not necessarily have any smooth low-dimensional or convex structure and in fact can be quite

"brittle"; in practice, the world model can be easily steered toward outputting any desired output state, as depicted in Figure 3. We address this in Section 3.3 with a reshaping of the descent directions.

We now describe our approach to address these two fundamental problems with lifted-states approaches to planning.

### 3.2. Exploration via Langevin state updates

The lifted optimization in (4) is still non-convex and can get trapped in poor local minima. In practice, we frequently observe that deterministic joint updates in $(\mathbf{a}, \mathbf{s})$ converge to "bad" stationary points where the intermediate variables settle into an unfavorable basin; for example, a linear route that ignores barriers or walls like in Figure 7. To circumvent this, we inject stochasticity directly into the state iterates, yielding a Langevin-style update that encourages exploration in the lifted state space.

**Langevin dynamics on state iterates.** Consider the optimization induced by (4). A standard way to escape spurious basins is to replace deterministic gradient descent on $\mathbf{s}$ with overdamped Langevin dynamics (Gelfand & Mitter, 1991), whose Euler discretization takes the following form:

$$s_t^{k+1} \leftarrow s_t^k - \eta_s \nabla_{s_t} \mathcal{L}_{\text{dyn}}(\mathbf{s}^k, \mathbf{a}^k) + \sigma_{\text{state}} \xi_t^k, \quad (5)$$

$$a_t^{k+1} \leftarrow a_t^k - \eta_a \nabla_{a_t} \mathcal{L}_{\text{dyn}}(\mathbf{s}^k, \mathbf{a}^k), \quad (6)$$

where $\xi_t^k \sim \mathcal{N}(0, I)$. That is, each optimization step performs a gradient descent update on the intermediate states, followed by an isotropic Gaussian perturbation. Intuitively, the noise allows the iterates to "hop" between nearby basins of the lifted loss landscape.

**Noise on states vs. actions.** By only noising the states, we can still condition on more dynamically feasible trajectories, while still allowing exploration over a wider distribution. Intuitively, planning problems often have a single (or small number of) intermediate states to find for the solution, and being able to noise directly over states rather than actions allows us to find these intermediate states faster. See Appendix A.3 for a characterization of the sampled distribution.

### 3.3. Sensitivity to state gradients

**A note on adversarial robustness of state gradients.** In practice, $F_\theta$ is learned and can have brittle local geometry. When optimizing (4) by gradient descent in both $\mathbf{a}$ and $\mathbf{s}$, we observed empirically that gradients with respect to the state inputs, $\nabla_s F_\theta(s, a)$, can be exploited: for any local goal-reaching objective of the following form:

$$\arg\min_s \|y - F_\theta(s, a)\|_2^2, \quad (7)$$

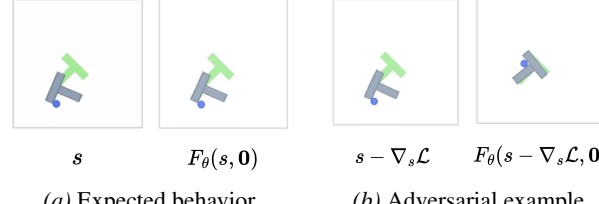

| $s$ | $F_\theta(s, \mathbf{0})$ | $s - \nabla_s \mathcal{L}$ | $F_\theta(s - \nabla_s \mathcal{L}, \mathbf{0})$ |

*(a)* Expected behavior      *(b)* Adversarial example

*Figure 3. Sensitivity of state gradient structure.* Examples of three states far away from the goal on the right (either in-distribution or out-of-distribution), such that taking a small step along the gradient $s' = s - \epsilon \nabla_s \mathcal{L}(s), \mathcal{L}(s) = \|F_\theta(s, a = \mathbf{0}) - g\|_2^2$, leads to a nearby state $s'$ that solves the planning problem in a single step: $F_\theta(s', \mathbf{0}) = g$. Thus, optimizing states directly through the world model $F_\theta$ can be quite challenging.

instead of the optimizer learning to find an $s$ on-manifold such that applying the action $a$ leads to end state $y$, the optimizer can find a nearby ambient $s + \delta, \|\delta\|_2 \ll 1$ such that the loss is practically minimized: $y \approx F_\theta(s + \delta, a)$, *regardless of the starting state $s$.* This is analogous to the adversarial robustness issue of image classifiers (Szegedy et al., 2013; Shamir et al., 2021): high-dimensional input spaces for trained neural networks can have high Lipschitz constants, hindering optimization performance.

Unfortunately, any loss function over $\mathbf{s}$ and $\mathbf{a}$ whose minimizers are feasible dynamics must depend on the state gradient $\nabla_s F$ in a meaningful way. We provide the informal theorem here, with formalization and proof in Appendix A.4.

**Theorem 3.1** (informal). *A differentiable loss function over state/action trajectories $\mathcal{L} : \mathcal{S}^T \times \mathcal{A}^T \to \mathbb{R}$ given a world model $F_\theta : \mathcal{S} \times \mathcal{A} \to \mathcal{S}$ cannot satisfy both of the following at the same time:*

1. *Minimizers of $\mathcal{L}$ correspond to dynamically feasible trajectories: $F_\theta(s_t, a_t) = s_{t+1}$,*

2. *$\mathcal{L}$ is insensitive to the world model state gradient $\nabla_s F_\theta$.*

To address this adversarial sensitivity, we detach gradients through the *state inputs* of the world model, while still differentiating with respect to the actions. We denote by $\bar{s}_t$ a stop-gradient copy of $s_t$ (i.e., $\bar{s}_t = s_t$ in value, but treated as constant during differentiation).

**Grad-cut dynamics loss.** We begin by applying a gradient stop to the state inputs in the dynamics loss:

$$\mathcal{L}_{\text{dyn}}^{\text{sg}}(\mathbf{s}, \mathbf{a}) = \sum_{t=0}^{T-1} \|F_\theta(\bar{s}_t, a_t) - s_{t+1}\|_2^2. \quad (8)$$

This objective is differentiable with respect to $\mathbf{a}$ and the *next* states $s_{t+1}$, but does not backpropagate through $s_t$ via $F_\theta$.

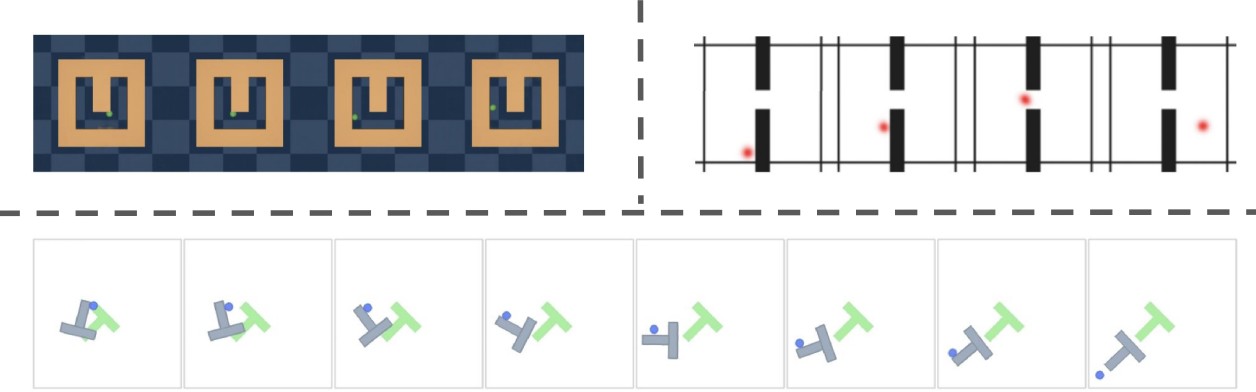

*Figure 4. Virtual states learned through planning.* All examples are instantiations of our planner at horizon 50 in the Point-Maze, Wall-Single, and Push-T environments. Regardless of the dynamics constraint relaxation and state noising, directly optimized states find realistic, non-greedy paths towards the goal.

**Dense goal shaping on one-step predictions.** While (8) improves robustness, it introduces a new degeneracy: paths gravitate towards the current rollout, regardless of proximity to the goal. To provide a task-aligned signal at every time step without state-input gradients, we add a goal loss on the one-step predictions:

$$\mathcal{L}_{\text{goal}}^{\text{sg}}(\mathbf{s}, \mathbf{a}) = \sum_{t=0}^{T-1} \left\| F_\theta(\bar{s}_t, a_t) - g \right\|_2^2. \quad (9)$$

This encourages each predicted next state to move toward the goal, supplying gradient information to every action $a_t$ while maintaining the grad-cut on $s_t$. This is depicted visually in Figure 2, and theoretically in Appendix A.4. Crucially, due to the stop-gradient $\bar{s}_t$, gradients through $F_\theta(\bar{s}_t, a_t)$ flow only with respect to $a_t$ (and not $s_t$), which prevents the optimizer from exploiting adversarial state-input directions. The final energy that is sampled from is then the following:

$$\begin{aligned}
\mathcal{L}(\mathbf{s}, \mathbf{a}) = &\sum_{t=0}^{T-1} \left\| F_\theta(\bar{s}_t, a_t) - s_{t+1} \right\|_2^2 \\
&+ \gamma \sum_{t=0}^{T-1} \left\| F_\theta(\bar{s}_t, a_t) - g \right\|_2^2,
\end{aligned} \quad (10)$$

where $\gamma > 0$ is fixed. While a dense goal loss would normally over-promote greedy trajectories, the stop-grad reshapes the optimization landscape in a way where non-greedy trajectories are still promoted, which we also observe in experimental results.

**Resulting noisy dynamics.** The final resulting dynamics,

after explicitly writing out $\nabla_a \mathcal{L}$, are as follows:

$$s_t^{k+1} \leftarrow s_t^k - 2\eta_s \Big( s_t - F_\theta(s_{t-1}, a_{t-1}) \Big) + \sigma_{\text{state}} \xi_t^k, \quad (11)$$

$$a_t^{k+1} \leftarrow a_t^k - \eta_a \nabla_{a_t} \mathcal{L}(\mathbf{s}^k, \mathbf{a}^k), \quad (12)$$

where $\xi_t^k \sim \mathcal{N}(0, I)$. Importantly, while the action dynamics still follow a gradient flow, the states do not follow a true gradient vector field, and thus *the resulting dynamics are not Langevin.* What results are still noisy dynamics that bias towards valid goal-oriented trajectories, but whose efficiency will require an extra synchronization step as described in the following section.

---

**Require:** Initial observation $o_0$, goal $o_g$, world model $F_\theta$, horizon $T$, steps $K$, learning rates $\eta_a, \eta_s$.
**Ensure:** $\mathbf{a}^*$
1: $s_0 \leftarrow \text{encode}(o_0)$, $s_T \leftarrow \text{encode}(o_g)$
2: $\mathbf{a} \leftarrow \mathbf{a}^0$; $\mathbf{s} \leftarrow \text{init\_states}(s_0, s_T)$
3: **for** $k = 0$ **to** $K - 1$ **do**
4:     Compute $\mathcal{L}$ as in (10)
5:     **Joint step:** $(\mathbf{a}, \mathbf{s}) \leftarrow (\mathbf{a}, \mathbf{s}) - (\eta_a \nabla_{\mathbf{a}} \mathcal{L}, \eta_s \nabla_{\mathbf{s}} \mathcal{L})$
6:     **Stochastic state:** $s_t \leftarrow s_t + \sigma_{\text{state}} \xi_t, \xi_t \sim \mathcal{N}(0, I)$ for $t = 1, \dots, T-1$
7:     **Sync (periodic):** if $k \bmod K_{\text{sync}} = 0$, rollout from $s_0$; $s_{t+1} \leftarrow F_\theta(s_t, a_t)$ for $t = 0, \dots, T-1$
8:         then take a GD step $\mathbf{a} \leftarrow \mathbf{a} - \eta_{\text{sync}} \nabla_{\mathbf{a}} \| s_T - g \|_2^2$
9: **end for**
10: **return** $\mathbf{a}^* \leftarrow \mathbf{a}$

---

*Algorithm 1.* GRASP planner: parallel stochastic gradient-based planning

### 3.4. Full-rollout synchronization

The no-state-gradient updates are designed to be robust to brittle state-input Jacobians of the learned world model. However, the stochastic optimization in (11) still needs a

method of strict descent towards true minima. In practice, we found it beneficial to periodically "sync" the plan by briefly running standard full-gradient planning on the original rollout objective.

**Full-gradient rollout step.** Every $K_{\text{sync}}$ iterations, we perform $J_{\text{sync}}$ steps of gradient descent on the original planning loss

$$\min_{\mathbf{a}} \|s_T(\mathbf{a}, s_0) - g\|_2^2, \tag{13}$$

where $s_T(\mathbf{a}, s_0)$ is computed by sequentially rolling out the world model

$$s_{t+1} = F_\theta(s_t, a_t), \qquad s_0 \text{ given.} \tag{14}$$

During this synchronization phase we update only the actions,

$$\mathbf{a} \leftarrow \mathbf{a} - \eta_{\text{sync}} \nabla_{\mathbf{a}} \|s_T(\mathbf{a}, s_0) - g\|_2^2, \tag{15}$$

using full backpropagation through the $T$-step rollout. By keeping these GD steps small relative to the stochastic dynamics of (11), we benefit from the smoothed loss landscape in Figure 1c for wider exploration, and the sharp but brittle landscape in Figure 1b for refinement.

# 4. Results

We evaluate our proposed planner GRASP across two complementary classes of environments designed to test (i) non-convex long-horizon planning with obstacles and (ii) data-driven visual control under learned dynamics. Concretely, these experiments aim to answer three questions:

1. Can the proposed planner overcome the greedy local minima that often trap shooting methods?

2. Does the method remain robust as the planning horizon increases?

3. Does the proposed planner converge to plans faster than rollout-based planners?

We provide self-ablations in Table 3, demonstrating the value of various components of our planner: using the state gradient detaching, the GD sync steps, and the level of the noise.

Figure 4 visualizes planning iterations in several navigation environments, illustrating how trajectories initialized far from dynamically consistent rollouts converge to feasible plans that satisfy the learned dynamics.

## 4.1. Baselines

We compare against three commonly used planners. **CEM** optimizes action sequences by iteratively sampling candidate trajectories, selecting elites, and refitting a sampling

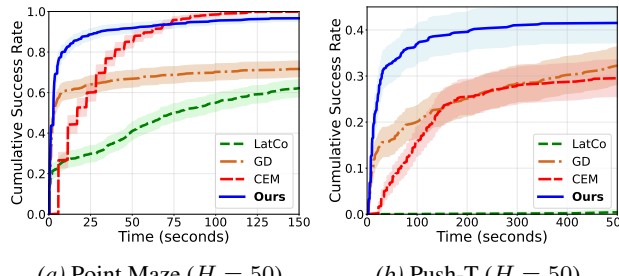

*(a)* Point Maze ($H = 50$)  *(b)* Push-T ($H = 50$)

*Figure 5.* **Success rate over time at a fixed horizon.** Success rate over fixed set of *open-loop* planning tasks for CEM, GD, LatCo (Rybkin et al., 2021), and our planner for a fixed horizon of 50. Curves summarize how quickly each planner makes progress under the learned world model setting when evaluated at a fixed planning horizon. Shaded regions are Wald 95% confidence intervals.

distribution. **GD** directly optimizes the action sequence by backpropagating through the dynamics model. **LatCo** (Rybkin et al., 2021) optimizes in a lifted latent/state-space by jointly adjusting intermediate latent variables and actions. This setting is different than the original LatCo method, which was originally applied in a model-based RL environment, but it still provides an important baseline for performance if we were to purely focus on direct optimization of (4).

For all methods, we sweep over hyperparameters and report results using the best-performing setting for each environment and horizon. For our planner, we initialize the states $\{s_t\}_{t=0}^T$ as noised around the linear interpolation between $s_0$ and $g$: $s_t = \frac{t}{T}g + (1 - \frac{t}{T})s_0 + z$, $z \sim \mathcal{N}(0, \epsilon I)$, and actions initialized at zeros: $a_t = 0$.

## 4.2. Environments and evaluation protocol

We evaluate planning on three visual control environments with learned dynamics: *PointMaze*, *Wall-Single*, and *Push-T*. World models are trained using the DINO-wm framework (Zhou et al., 2024), following the original paper's setup, where the world model $F_\theta(s, a)$ takes 5 actions and predicts 5 steps ahead; that is, if $\dim(\mathcal{A}) = 2$, then $F_\theta$ takes actions as vectors of stacked actions $\mathbf{a} \in \mathbb{R}^{10}$. All world models predict 5 states ahead, and reported horizons are environment steps (so horizon of 80 corresponds to 16 world model evaluations). Hyperparameters can be found in Appendix C as well as the project code. The same hyperparameters were used for each planner across all DINO-WM environments.

All reported metrics measure task success under the learned world model. Success is defined as reaching the goal region within the planning horizon when actions are applied *in the simulated environment* (as opposed to the world model's predicted state). Success rates are reported as *best-by*: that

| | $H$=40 | $H$=50 | $H$=60 | $H$=70 | $H$=80 |
|---|---|---|---|---|---|
| | Succ. / Time | Succ. / Time | Succ. / Time | Succ. / Time | Succ. / Time |
| CEM | **61.4** / 35.3s | 30.2 / 96.2s | 7.2 / 83.1s | 7.8 / 156.1s | 2.8 / 132.2s |
| LatCo | 15.0 / 598.0s | 4.2 / 1114.7s | 2.0 / 231.5s | 0.0 / — | 0.0 / — |
| GD | 51.0 / 18.0s | 37.6 / 76.3s | 16.4 / 146.5s | 12.0 / 103.1s | 6.4 / 161.3s |
| GRASP (Ours) | 59.0 / **8.5s** | **43.4 / 15.2s** | **26.2 / 49.1s** | **16.0 / 79.9s** | **10.4 / 58.9s** |

*Table 1.* Open-loop planning results on long range Push-T. Reported are success rate (%) and median success time (seconds; successful trials only) across planning horizons. 500 trials per setting. Each cell reports *Success / Time*.

*Table 2.* Additional experiments on le-wm and jepa-wm. Daggers correspond to minor modifications to GRASP; single dagger corresponds to replacing the GD sync step with a small CEM step, and double dagger corresponds to linearly decaying both $\sigma_{\text{state}}$ and the goal loss coefficient $\lambda$ between each sync step.

| Environment | Horizon | CEM | GD | GRASP |
|---|---|---|---|---|
| *le-wm* | | | | |
| Reacher | 75 | 38% | 42% | **70%** |
| PushT | 75 | 22% | 26% | **46%**[†] |
| Cube | 50 | 58% | 56% | **66%**[‡] |
| *jepa-wm* | | | | |
| Metaworld | 30 | 12% | 6% | **19%**[‡] |

*Table 3.* Ablation Studies over the GD sync steps, level of noise for Langevin dynamics, and whether we use our detached gradient approach with the goal-reaching objective or not. GD sync happens every 100 stochastic steps. Ablations done on the Push-T environment at horizon $H = 40$. Time reported is median over successful trials, which only beats our method when the accuracy is much smaller.

| Setting | Accuracy (%) | Time (s) |
|---|---|---|
| *Sync Steps* | | |
| No Sync | 1.6 | 4.5 |
| 10 | 48.0 | 11.2 |
| 25 (ours) | **59.0** | 8.5 |
| 50 | 58.0 | 9.8 |
| *Noise Level* | | |
| $\sigma = 0.0$ | 54.8 | 10.4 |
| $\sigma = 0.5$ (ours) | **59.0** | 8.5 |
| $\sigma = 1.0$ | 50.6 | 8.4 |
| *Detach $\nabla_s F(s, a)$* | | |
| Stop (ours) | **59.0** | 8.5 |
| Flow | 46.6 | 10.3 |

is, out of all trials, the number of trials that successfully reached the goal at some point before the maximum allotted time of 30 minutes.

### 4.3. Long-term planning and horizon scaling

We evaluate planners in the long-horizon regime, where our parallelized stochastic planner is more intended; greedy local minima and optimization instability become the dominant challenges.

GRASP remains reliable as the planning horizon increases: it solves more tasks and finishes a majority of its successful trials faster, showing stronger robustness to long horizons than the baselines as shown in Table 1. Beyond the median completion times, we provide further illustration of the solving speed of our planner in Figure 5, showing further that most of its plans converge at an earlier time. At the longer horizons for Push-T, where it is more important for the planner to explore non-greedy optima, GRASP finds more success than the baselines and is able to find the needed non-greedy trajectories, as visualized in Figure 6.

### 4.4. Additional world models

To demonstrate the flexibility of GRASP to be deployed to other more recent world models, we also present experiments on *le-wm* (Maes et al., 2026) and *jepa-wm* (Terver et al., 2025), with success rates given in Table 2. On the reacher environment for le-wm, the GRASP planner needed

minimal tuning to perform well, only needing to tune the action learning rate from the baseline hyperparameters. For the le-wm on push-t, we found it helpful to replace the GD sync step with a small CEM sync step. Finally, for le-wm's cube and jepa-wm's metaworld-reach, we found it helpful to linearlly decay both the state noise variance $\sigma_{\text{state}}$ and the goal loss coefficient $\gamma$ between each sync step (and reset right after each sync step). Implementations for GRASP in both jepa-wm and le-wm are provided from our main code repository[1].

### 4.5. Short-term planning

We also evaluate short-horizon planning, to demonstrate that our planner can match performance on shorter, easier tasks. Table 4 reports success rates across environments for horizons ranging from $H = 10$ to $H = 30$, while Table 5 reports median wall-clock planning times.

Across all environments and short horizons, the proposed

---
[1]https://github.com/michael-psenka/grasp

| Planner | Push-T | | | PointMaze | | | WallSingle | | |
|---|---|---|---|---|---|---|---|---|---|
| | $H{=}10$ | $H{=}20$ | $H{=}30$ | $H{=}10$ | $H{=}20$ | $H{=}30$ | $H{=}10$ | $H{=}20$ | $H{=}30$ |
| CEM | **100.0** | **96.0** | **76.0** | **100.0** | **100.0** | **100.0** | **98.0** | **100.0** | **100.0** |
| LatCo | 67.4 | 55.2 | 35.4 | **100.0** | 99.8 | 99.0 | 91.4 | 81.6 | 62.8 |
| GD | 99.2 | 88.4 | 69.0 | 94.6 | 91.8 | 94.6 | 91.8 | 92.8 | 91.6 |
| GRASP (Ours) | **100.0** | 89.2 | 75.2 | **100.0** | **100.0** | 99.8 | 95.2 | 91.8 | 95.0 |

*Table 4.* Short Term Planning. Success rate (%) for Push-T, PointMaze, and WallSingle. 500 trials per setting. Our method has comparable success rate while having a consistently low completion time (Table 5).

| Planner | Push-T | | | PointMaze | | | WallSingle | | |
|---|---|---|---|---|---|---|---|---|---|
| | $H{=}10$ | $H{=}20$ | $H{=}30$ | $H{=}10$ | $H{=}20$ | $H{=}30$ | $H{=}10$ | $H{=}20$ | $H{=}30$ |
| CEM | 1.3 | 7.5 | 23.6 | 0.7 | 2.0 | 6.3 | 0.9 | 1.5 | 2.1 |
| LatCo | 39.9 | 437.2 | 1800.0 | 1.6 | 13.7 | 36.6 | 12.1 | 23.8 | 111.1 |
| GD | **0.5** | 2.4 | 25.3 | **0.3** | **0.6** | **1.2** | **0.4** | **0.5** | **0.7** |
| GRASP (Ours) | 1.5 | **2.1** | **9.1** | 0.7 | 1.8 | 2.1 | 1.6 | 1.8 | 2.1 |

*Table 5.* Median completion times (seconds) across short-term experiments. 500 trials per setting. Our method has a consistently low completion time with a comparable success rate (Table 4).

planner achieves success rates comparable to the baselines. Alongside similar success rates, our proposed planner exhibits consistently low planning times. As shown in Table 5, it is among the fastest methods across all environments and horizons, often significantly faster than sampling-based approaches and competitive with gradient-based optimization, sacrificing some speed for a higher success rate. These results indicate that even in relatively short and easy planning regimes, our planner remains competitive with the baselines.

Overall, these results demonstrate that GRASP consistently matches strong baselines in short-term planning, while outperforming them in long-horizon settings by avoiding greedy failures and converging more quickly in wall-clock time.

## 5. Related work

**World modeling** has shown significant improvement in sample efficiency for model-based reinforcement learning (Hafner et al., 2025). By learning to predict future states given current states and actions, world models enable planning without access to an interactive environment (Ding et al., 2024). Recent work has focused on learning latent-space representations to handle high-dimensional observations (Assran et al., 2025), with models now demonstrating the ability to scale and generalize across diverse environments (Bar et al., 2025). In this paper, we develop an efficient planning algorithm for action-conditioned video models.

**Sampling-based planning in world models** traditionally relies on methods like the Cross-Entropy Method (CEM (Rubinstein & Kroese, 2004)) and random shooting. While

these methods are robust and simple to implement, they suffer from serial evaluation bottlenecks and poor scaling with planning horizon length (Bharadhwaj et al., 2020). Recent work proposes performance improvements—such as faster CEM variants with action correlation and memory, parallelized sampling via diffeomorphic transforms, and massively parallel strategies—but fundamental limitations remain for very long horizons (Pinneri et al., 2021; Lai et al., 2022).

**Gradient-based planning** leverages the differentiability of neural world models to optimize action sequences directly (Jyothir et al., 2023). Early approaches applied backpropagation through time to optimize actions (Thrun et al., 1990), but face challenges with vanishing/exploding gradients and poor conditioning over long horizons. Hybrid strategies combining gradient descent with sampling-based methods—such as interleaving CEM with gradient updates—have shown promise. CEM-GD variants interleave backwards passes through the learned model with population-based search for improved convergence and scalability (Bharadhwaj et al., 2020; Huang et al., 2021). Recent work improves gradient-based planners by training world models to be adversarially robust to improve gradient-based planning (Parthasarathy et al., 2025); our method instead tries to improve gradient-based planning for more general world models, without any pretraining modifications.

**State optimization and multiple shooting in optimal control.** The idea of treating states as optimization variables separate from dynamics constraints has a rich history in classical optimal control. Non-condensed QP formulations in MPC (Jerez et al., 2011) decouple state and input optimization for improved numerical properties. Multiple shooting

methods (Tamimi & Li, 2009; Diedam & Sager, 2018) break long-horizon problems into shorter segments with continuity constraints. Direct collocation approaches (Bordalba et al., 2022; Nie & Kerrigan, 2025) optimize state and control trajectories simultaneously while enforcing dynamics through collocation constraints. These methods have primarily been applied to systems with known analytical dynamics. Trajectory optimization methods in robotics have developed parallel shooting techniques and GPU-accelerated planning algorithms (Guhathakurta et al., 2022), but most approaches still face fundamental limitations when applied to learned world models, particularly visual world models where dynamics are approximate and high-dimensional. One parallel work explores a collocation-based approach for grounding action-conditioned video models on real trajectories (Ziakas et al., 2026) but is focused more on trajectory grounding than planning, for example not incorporating a stochastic component for exploration.

**Noise and regularization in optimization.** Stochastic optimization techniques and noise injection have long been recognized for their ability to improve optimization outcomes (Robbins & Monro, 1951), and can help regularize and explore complex loss landscapes (Welling & Teh, 2011; Xu et al., 2018; Bras, 2023; Foret et al., 2020). In the context of planning, noise is commonly used in sampling-based methods, but its systematic incorporation into gradient-based planning for learned models remains underexplored.

## 6. Limitations and future work

Although the proposed planner shows clear advantages in long-horizon settings, its benefits are more limited at short horizons. As demonstrated in our experiments, for small planning horizons the planner typically achieves success rates and completion times that are comparable to strong baselines such as CEM and gradient-based optimization, rather than strictly outperforming them. This suggests that the primary gains of the method arise in regimes where long-horizon reasoning and non-greedy planning are essential, rather than in short-horizon settings where simpler methods already perform well.

Hybrid planners (that combine iterations of a rollout-based planner like CEM and a gradient-based planner like GD) have been implemented to get the "best of both worlds" from the two approaches (Huang et al., 2021; Nachkov et al., 2026), and there are many ways to "hybridize" our planner as well. We leave exploration of such methods for future work.

Several components of the planner are designed to mitigate the unreliability of state gradients in learned world models. While effective, these modifications introduce additional structure and hyperparameters that would ideally be un-

necessary. If state representations induced by the world model were smoother or more geometrically well-behaved in the state space, many of these stabilization mechanisms could be removed, potentially leading to further speed improvements. Promising directions toward this goal include improved representation learning through adversarial training, diffusion-based world models, or other techniques that explicitly regularize the geometry of the learned state space.

## 7. Conclusion

World models provide a powerful framework for planning in complex environments, but existing approaches struggle with long horizons, high-dimensional actions, and serial computation. We propose GRASP, a new gradient-based planning algorithm with two key contributions: (a) a lifted planner that optimizes actions together with "virtual states" in a time-parallel manner, yielding more stable and scalable optimization while allowing direct control over exploration via stochastic state updates, and (b) an action-gradient-only planning variant for learned visual world models that avoids brittle state-input gradients while still exploiting differentiability with respect to actions. Experiments on visual world-model benchmarks show that our approach remains robust as horizons grow and finds non-greedy solutions at a faster rate than commonly used planners such as CEM or vanilla GD.

## Impact Statement

This paper presents work whose goal is to advance the field of Machine Learning. There are many potential societal consequences of our work, none which we feel must be specifically highlighted here.

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

# A. Theory

In this section, we provide a theoretical analysis of the convergence properties of our planning approach compared to traditional shooting methods. We consider a simplified linear dynamics setting to derive formal convergence guarantees. We reproduce proof here for self-containedness; see e.g. (Ascher et al., 1995) for other theoretical treatments of this problem.

## A.1. Convergence of various methods in the convex setting

Consider a linear dynamical system with dynamics:

$$s_{t+1} = As_t + Ba_t, \tag{16}$$

where $s_t \in \mathbb{R}^n$ is the state at time $t$, $a_t \in \mathbb{R}^m$ is the control input, and $A \in \mathbb{R}^{n \times n}$, $B \in \mathbb{R}^{n \times m}$.

Let $\mathcal{F}(s_0, \mathbf{a}) : \mathbb{R}^n \times \mathbb{R}^{mT} \to \mathbb{R}^n$ denote the rollout of the dynamics for $T$ timesteps. Under linear dynamics, $\mathcal{F}$ takes the compact form:

$$\mathcal{F}(s_0, \mathbf{a}) = A^T s_0 + \sum_{t=0}^{T-1} A^{T-1-t} Ba_t. \tag{17}$$

We analyze the optimization landscape of two fundamental formulations for reaching a target state $g$ from initial state $s_{init}$.

The *Shooting Method* minimizes the error at the final step relative to the control inputs $\mathbf{a}$, implicitly simulating the dynamics:

$$\min_{\mathbf{a} \in \mathbb{R}^{mT}} J_S(\mathbf{a}) = \|\mathcal{F}(s_{init}, \mathbf{a}) - g\|_2^2 \tag{18}$$

The *Lifted States Method* (or Multiple Shooting) treats both states and controls as optimization variables and minimizes the violation of the dynamics constraints (the physics defects):

$$\min_{\mathbf{a}, \mathbf{s}} \quad J_L(\mathbf{a}, \mathbf{s}) = \sum_{t=0}^{T-1} \|As_t + Ba_t - s_{t+1}\|_2^2 \tag{19}$$

$$\text{subject to} \quad s_0 = s_{init}, \quad s_T = g \tag{20}$$

where $\mathbf{s} = (s_0, \ldots, s_T)$.

**Matrix Representation** To analyze the convergence properties, we express both objectives in quadratic matrix form.

**Shooting Method.** Let $\mathbf{a} = (a_0; \ldots; a_{T-1}) \in \mathbb{R}^{mT}$. The final state is linear in $\mathbf{a}$:

$$s_T = A^T s_{init} + \mathcal{C}_T \mathbf{a} \tag{21}$$

where $\mathcal{C}_T = [A^{T-1}B, A^{T-2}B, \ldots, B] \in \mathbb{R}^{n \times mT}$ is the controllability matrix. The objective is:

$$J_S(\mathbf{a}) = \|\mathcal{C}_T \mathbf{a} - (g - A^T s_{init})\|_2^2 \tag{22}$$

The Hessian of this objective is $H_S = 2\mathcal{C}_T^\top \mathcal{C}_T$.

**Lifted Method.** We eliminate the fixed boundary variables $s_0$ and $s_T$ and optimize over the free variables $\mathbf{z} = (s_1; \ldots; s_{T-1}; a_0; \ldots; a_{T-1})$. The dynamics residuals can be written as a linear system $M\mathbf{z} - b$. The dynamics equations for $t = 0, \ldots, T-1$ correspond to rows in a large matrix $M$.

- For $t = 0$: $s_1 - Ba_0 = As_{init}$ (Const. term moved to RHS)

- For $0 < t < T-1$: $As_t + Ba_t - s_{t+1} = 0$

- For $t = T-1$: $-As_{T-1} - Ba_{T-1} = -g$ ($s_T$ fixed to $g$)

The objective is $J_L(\mathbf{z}) = \|M\mathbf{z} - b\|_2^2$, and the Hessian is $H_L = 2M^\top M$. The matrix $M$ has a sparse, block-banded structure.

**Smoothness Analysis** We compare the smoothness of the two optimization problems by comparing the Lipschitz constants of their gradients, $L = \lambda_{\max}(H)$.

**Theorem A.1** (Shooting: Exploding Smoothness). *Let $A^\top$ have a real eigenvalue $\lambda$ with $|\lambda| > 1$ and unit eigenvector $v$ (a left eigenvector of $A$). Assume $B$ aligns with this mode such that for some input direction $w$ ($\|w\|_2 = 1$), the projection $|\langle v, Bw \rangle| = \mu > 0$.*

*Then, the Lipschitz constant of the Shooting method gradient grows exponentially with $T$:*

$$L_S = \lambda_{\max}(H_S) \geq 2\mu^2 |\lambda|^{2(T-1)}. \tag{23}$$

*Proof.* The Lipschitz constant is the maximum eigenvalue of the Hessian $H_S = 2\mathcal{C}_T^\top \mathcal{C}_T$, which equals $2\|\mathcal{C}_T\|_2^2$. By definition, the spectral norm is the maximum gain over all possible inputs:

$$\|\mathcal{C}_T\|_2 = \sup_{\mathbf{a} \neq 0} \frac{\|\mathcal{C}_T \mathbf{a}\|_2}{\|\mathbf{a}\|_2}. \tag{24}$$

From the existence of a controllable non-contractive mode, we can construct a unit-norm $\mathbf{a}_{test} = (w; \mathbf{0}; \ldots; \mathbf{0})$ that evaluates to the form:

$$\mathcal{C}_T \mathbf{a}_{test} = A^{T-1} Bw, \tag{25}$$

and such that the following holds when projecting to the corresponding unstable eigenvector $v$:

$$\|\mathcal{C}_T \mathbf{a}_{test}\|_2 \geq |\langle v, A^{T-1} Bw \rangle|, \tag{26}$$
$$= |\langle (A^\top)^{T-1} v, Bw \rangle|, \tag{27}$$
$$= |\langle \lambda^{T-1} v, Bw \rangle|, \tag{28}$$
$$= |\lambda|^{T-1} |\langle v, Bw \rangle|, \tag{29}$$
$$= \mu |\lambda|^{T-1}. \tag{30}$$

Squaring this result gives the desired bound. $\square$

**Theorem A.2** (Lifted: Stable Smoothness). *The Lipschitz constant of the Lifted gradient is bounded by a constant independent of the horizon $T$. Specifically, one valid bound is:*

$$L_L = \lambda_{\max}(H_L) \leq 6\big(1 + \|A\|_2^2 + \|B\|_2^2\big). \tag{31}$$

*Proof.* The Hessian is $H_L = 2M^\top M$, so $\lambda_{\max}(H_L) = 2\|M\|_2^2$. It therefore suffices to upper bound $\|M\|_2$ by a constant that does not depend on $T$.

The matrix $M$ is block-sparse: each block row corresponding to timestep $t$ contains at most three non-zero blocks, namely an identity block (selecting $s_{t+1}$), a dynamics block (multiplying $s_t$), and an input block (multiplying $a_t$). Equivalently, the corresponding residual has the form

$$r_t = As_t + Ba_t - s_{t+1}, \tag{32}$$

with $s_0$ and $s_T$ treated as fixed boundary values (so $r_t$ is affine in the free variables).

Fix any optimization vector $\mathbf{z} = (s_1; \ldots; s_{T-1}; a_0; \ldots; a_{T-1})$ (stacking the free states and controls), and let $M\mathbf{z}$ denote the stacked residuals $(r_0, \ldots, r_{T-1})$ with constants removed. Using the inequality $\|x + y + z\|_2^2 \leq 3(\|x\|_2^2 + \|y\|_2^2 + \|z\|_2^2)$ and the operator norm bounds $\|As\|_2 \leq \|A\|_2 \|s\|_2$, $\|Ba\|_2 \leq \|B\|_2 \|a\|_2$, we obtain for each $t$:

$$\|r_t\|_2^2 = \| - s_{t+1} + As_t + Ba_t \|_2^2 \tag{33}$$
$$\leq 3\Big( \|s_{t+1}\|_2^2 + \|A\|_2^2 \|s_t\|_2^2 + \|B\|_2^2 \|a_t\|_2^2 \Big). \tag{34}$$

Summing over $t = 0, \ldots, T-1$ gives

$$\|M\mathbf{z}\|_2^2 = \sum_{t=0}^{T-1} \|r_t\|_2^2 \leq 3\sum_{t=0}^{T-1}\Big(\|s_{t+1}\|_2^2 + \|A\|_2^2\|s_t\|_2^2 + \|B\|_2^2\|a_t\|_2^2\Big). \tag{35}$$

Crucially, due to the banded structure, each free intermediate state $s_1, \ldots, s_{T-1}$ appears in at most two terms in the sum: once as $s_{t+1}$ and once as $s_t$. Hence the state contributions can be bounded without any accumulation in $T$, yielding the following:

$$\|M\mathbf{z}\|_2^2 \leq 3\Big((1 + \|A\|_2^2)\sum_{t=1}^{T-1}\|s_t\|_2^2 + \|B\|_2^2\sum_{t=0}^{T-1}\|a_t\|_2^2\Big), \tag{36}$$

$$\leq 3\big(1 + \|A\|_2^2 + \|B\|_2^2\big)\,\|\mathbf{z}\|_2^2. \tag{37}$$

Therefore, $\|M\|_2^2 = \sup_{\mathbf{z}\neq 0}\|M\mathbf{z}\|_2^2/\|\mathbf{z}\|_2^2 \leq 3(1 + \|A\|_2^2 + \|B\|_2^2)$, and thus

$$L_L = 2\|M\|_2^2 \leq 6\big(1 + \|A\|_2^2 + \|B\|_2^2\big), \tag{38}$$

which is independent of $T$. $\qquad\square$

**Interpretation.** While we needed a slightly more restrictive assumption for the lower bound on the Shooting method's Hessian, such a condition is not unreasonable to find, as many realistic systems will not be universally stable within the controllability subspace. In these settings, the shooting method forces the optimization to traverse a loss landscape with curvature that varies exponentially (requiring exponentially small steps to remain stable), while the lifted states method "preconditions" the problem by creating variables for intermediate states, decoupling the long-term dependencies into local constraints. This results in a loss landscape with uniform smoothness $O(1)$ with respect to the planning horizon length $T$.

## A.2. Gaussian noise regularization

The regularity from noisy gradient descent (or Langevin-based optimization) primarily stems from the smoothing of the Gaussian convolution:

**Theorem A.3** (Gaussian smoothing contracts gradients and yields scale control)**.** *Let $d \geq 1$, $\sigma > 0$, and let*

$$\phi_\sigma(\boldsymbol{\epsilon}) = (2\pi\sigma^2)^{-d/2}\exp\big(-\|\boldsymbol{\epsilon}\|^2/(2\sigma^2)\big) \tag{39}$$

*be the density of $\mathcal{N}(0, \sigma^2\mathbf{I}_d)$. For $L \in \mathcal{C}^1(\mathbb{R}^d)$ define*

$$L_\sigma(\mathbf{s}) = (\phi_\sigma * L)(\mathbf{s}) = \int_{\mathbb{R}^d}\phi_\sigma(\boldsymbol{\epsilon})\,L(\mathbf{s} - \boldsymbol{\epsilon})\,d\boldsymbol{\epsilon}. \tag{40}$$

*The following statements hold of the resulting convolution:*

1. *(Regularity never decreases.) For any $1 \leq p \leq \infty$, if the distributional gradient $\nabla_s L \in L^p(\mathbb{R}^d)$, then*

$$\|\nabla_s L_\sigma\|_{L^p} \leq \|\nabla_s L\|_{L^p}. \tag{41}$$

   *In particular, if $L$ is Lipschitz, then $\mathrm{Lip}(L_\sigma) = \|\nabla_s L_\sigma\|_{L^\infty} \leq \|\nabla_s L\|_{L^\infty} = \mathrm{Lip}(L)$.*

2. *(Explicit regularity control by variance.) For any $1 \leq p \leq \infty$, if $L \in L^p(\mathbb{R}^d)$, then*

$$\|\nabla_a L_\sigma\|_{L^p} \leq \frac{c_d}{\sigma}\|L\|_{L^p}, \qquad c_d := \mathbb{E}\|Z\| = \sqrt{2}\,\frac{\Gamma\big(\frac{d+1}{2}\big)}{\Gamma\big(\frac{d}{2}\big)}, \;\; Z \sim \mathcal{N}(0, \mathbf{I}_d). \tag{42}$$

   *Applying $p = \infty$, we get $\mathrm{Lip}(L_\sigma) \leq \frac{c_d}{\sigma}\|L\|_{L^\infty}$.*

*Proof.* An important property of convolution is that gradients commute to either argument:

$$\nabla_s L_\sigma = \nabla_s(\phi_\sigma * L), \tag{43}$$
$$= (\nabla\phi_\sigma) * L, \tag{44}$$
$$= \phi_\sigma * (\nabla_s L), \tag{45}$$

where $\nabla\phi_\sigma(\epsilon) = -(\epsilon/\sigma^2)\,\phi_\sigma(\epsilon)$.

For part 1, apply Young's convolution inequality with the fact that $\|\phi_\sigma\|_{L^1} = 1$. For any $g \in L^p$,

$$\|g * \phi_\sigma\|_{L^p} \le \|g\|_{L^p}. \tag{46}$$

Taking $g = \nabla_a L$ and using the commutation identity above gives

$$\|\nabla_s L_\sigma\|_{L^p} = \|\phi_\sigma * (\nabla_s L)\|_{L^p} \le \|\nabla_s L\|_{L^p}. \tag{47}$$

When $p = \infty$, $\|\nabla_s L\|_{L^\infty}$ is the Lipschitz constant of $L$, so $\mathrm{Lip}(L_\sigma) \le \mathrm{Lip}(L)$.

For part 2, again use the commutation identity together with Young's inequality in the form $\|h * k\|_{L^p} \le \|h\|_{L^1}\|k\|_{L^p}$. We obtain

$$\|\nabla_a L_\sigma\|_{L^p} = \|(\nabla\phi_\sigma) * L\|_{L^p} \le \|\nabla\phi_\sigma\|_{L^1}\|L\|_{L^p}. \tag{48}$$

It remains to compute $\|\nabla\phi_\sigma\|_{L^1}$. Since $\|\nabla\phi_\sigma(\epsilon)\|_2 = \|\epsilon\|_2\phi_\sigma(\epsilon)/\sigma^2$ and $X \sim \mathcal{N}(0, \sigma^2\mathbf{I}_d)$ has density $\phi_\sigma$, we get

$$\|\nabla\phi_\sigma\|_{L^1} = \int_{\mathbb{R}^d} \frac{\|\epsilon\|_2}{\sigma^2}\,\phi_\sigma(\epsilon)\,d\epsilon \tag{49}$$
$$= \frac{1}{\sigma^2}\,\mathbb{E}\|X\|_2 \tag{50}$$
$$= \frac{1}{\sigma}\,\mathbb{E}\|Z\|_2 \tag{51}$$
$$= \frac{c_d}{\sigma}, \tag{52}$$

where $X = \sigma Z$ with $Z \sim \mathcal{N}(0, \mathbf{I}_d)$. Substituting this into the previous display yields the claimed bound. The $p = \infty$ statement is the corresponding Lipschitz estimate. $\square$

We then get regularity in expectation by noting that, after adding noise to a gradient step, this noise is then fed as input into the next step.

**Corollary A.4** (Noise induces smoothed gradients in expectation). *Let $\boldsymbol{\xi} \sim \mathcal{N}(0, \sigma^2\mathbf{I}_d)$ and define the Gaussian-smoothed loss*

$$L_\sigma(\mathbf{s}) := \mathbb{E}_{\boldsymbol{\xi}}\big[L(\mathbf{s} + \boldsymbol{\xi})\big]. \tag{53}$$

*Assume $L \in \mathcal{C}^1(\mathbb{R}^d)$ and that $\mathbb{E}_{\boldsymbol{\xi}}\|\nabla L(\mathbf{s} + \boldsymbol{\xi})\| < \infty$ (so differentiation may be interchanged with expectation). Then*

$$\mathbb{E}_{\boldsymbol{\xi}}\big[\nabla L(\mathbf{s} + \boldsymbol{\xi})\big] = \nabla_{\mathbf{s}} L_\sigma(\mathbf{s}). \tag{54}$$

*Moreover, if $L \in L^\infty(\mathbb{R}^d)$, then by Theorem A.3 (Part 2),*

$$\|\nabla_{\mathbf{s}} L_\sigma\|_{L^\infty} \le \frac{c_d}{\sigma}\,\|L\|_{L^\infty}, \qquad \text{and hence} \qquad \left\|\mathbb{E}_{\boldsymbol{\xi}}\big[\nabla L(\mathbf{s} + \boldsymbol{\xi})\big]\right\| \le \frac{c_d}{\sigma}\,\|L\|_{L^\infty}. \tag{55}$$

This then provides motivation for the decaying noise schedule: as an annealing process from a smoother, less accurate gradient structure to a rigid but more accurate gradient structure. For more detailed theory on the regularity of noisy gradient descent, see e.g. (Chaudhari et al., 2019) and the references therein.

### A.3. Connection to Boltzmann sampling, state-only noising

The continuous-time Langevin dynamics corresponding to (5), if we were to also noise the actions $a_t$ similarly, has a stationary distribution that concentrates on low-energy regions of $\mathcal{L}_{\text{dyn}}(\mathbf{s}, \mathbf{a})$; in particular, under mild regularity conditions it admits the Gibbs (Boltzmann) density

$$p(\mathbf{s}, \mathbf{a}) \propto \exp\left(-\beta\, \mathcal{L}_{\text{dyn}}(\mathbf{s}, \mathbf{a})\right), \tag{56}$$

for an inverse temperature $\beta > 0$ determined by the relative scaling between the drift and diffusion terms[2]. However, we are only noising the states, so the converged distribution is not Boltzmann. The new converged distribution, written loosely, collapses on solutions in action to the local dynamics problems:

$$p(\mathbf{s}, \mathbf{a}) \propto \delta(\mathbf{a} = \mathbf{a}^*(\mathbf{s})) \cdot \exp\left(-\beta\, \mathcal{L}_{\text{dyn}}(\mathbf{s}, \mathbf{a})\right), \tag{57}$$

where $\mathbf{a}^*(\mathbf{s}) = \min_{\mathbf{a}} \mathcal{L}_{\text{dyn}}(\mathbf{s}, \mathbf{a})$.

### A.4. State-gradient-free Dynamics

We now analyze a variant of the optimization method where we "cut" the gradient flow through the state evolution in the dynamics. This is akin to removing the adjoint (backward) pass through time, transforming the global trajectory optimization into a sequence of locally regularized problems.

First, we provide proof that there are no loss functions where (i) the minimizers correspond to dynamically feasible trajectories, that also (ii) has no dependency on the state gradients through the world model $F_\theta(s, a)$. We formalize this in the following theorem:

**Theorem A.5** (Nonexistence of exact dynamics-enforcing losses with Jacobian-free state gradients). *Let $\mathcal{S} \subseteq \mathbb{R}^n$ and $\mathcal{A} \subseteq \mathbb{R}^m$ each be open and connected sets, and fix $s_0 \in \mathcal{S}$ and $g \in \mathcal{S}$. Consider horizon $T = 2$ with decision variables $(s_1, a_0, a_1) \in \mathcal{S} \times \mathcal{A} \times \mathcal{A}$ and boundary $s_2 = g$ fixed. Let $L_F : \mathcal{S} \times \mathcal{A} \times \mathcal{A} \to \mathbb{R}$ be decomposable into the following form:*

$$L_F(s_1, a_0, a_1) = \Phi(s_1, a_0, a_1, y_0, y_1), \tag{58}$$
$$y_0 = F(s_0, a_0), \tag{59}$$
$$y_1 = F(s_1, a_1), \tag{60}$$

*where $\Phi : \mathcal{S} \times \mathcal{A} \times \mathcal{A} \times \mathcal{S} \times \mathcal{S} \to \mathbb{R}$ is $C^1$. Let*

$$\mathcal{M}(F) = \{(s_1, a_0, a_1) : s_1 = F(s_0, a_0) \text{ and } g = F(s_1, a_1)\} \tag{61}$$

*denote the set of trajectories that satisfy the dynamics exactly at both steps (with the fixed boundary conditions). There does not exist such a $\Phi$ for which the following two properties hold simultaneously for every $C^1$ model $F$:*

1. *Minimizers correspond to feasible dynamics:* $\arg\min_{(s_1, a_0, a_1)} L_F(s_1, a_0, a_1) = \mathcal{M}(F)$,

2. *Independence of loss to the dynamics' state gradient: for all $G : \mathcal{S} \times \mathcal{A} \to \mathcal{S}, G \in C^1, \left(F(s_0, a_0) = G(s_0, a_0), F(s_1, a_1) = G(s_1, a_1)\right) \Rightarrow \nabla_{s_1} L_F(s_1, a_0, a_1) = \nabla_{s_1} L_G(s_1, a_0, a_1).$*

*Proof.* We argue by contradiction.

Fix any point $(s_1, a_0, a_1)$ and write $y_0 = F(s_0, a_0)$ and $y_1 = F(s_1, a_1)$. By the chain rule,

$$\nabla_{s_1} L_F(s_1, a_0, a_1) = \frac{\partial \Phi}{\partial s_1}(s_1, a_0, a_1, y_0, y_1) + \left(\nabla_s F(s_1, a_1)\right)^\top \frac{\partial \Phi}{\partial y_1}(s_1, a_0, a_1, y_0, y_1). \tag{62}$$

We claim that $\partial \Phi / \partial y_1$ must vanish identically. To see this, fix arbitrary arguments $(s_1, a_0, a_1, y_0, y_1), (s_0, a_0) \neq (s_1, a_1)$ in the domain and construct two $C^1$ models $F$ and $G$ such that $F(s_0, a_0) = G(s_0, a_0) = y_0$ and $F(s_1, a_1) = G(s_1, a_1) = y_1$, but whose state Jacobians at $(s_1, a_1)$ are prescribed arbitrarily and differ:

$$\nabla_s F(s_1, a_1) = J_F, \qquad \nabla_s G(s_1, a_1) = J_G. \tag{63}$$

---

[2]Equivalently, one can view $\sigma_{\text{state}}$ in (5) as setting an effective temperature: larger $\sigma_{\text{state}}$ yields broader exploration, while smaller $\sigma_{\text{state}}$ concentrates around local minima.

To construct, choose a small ball $B$ around $(s_1, a_1)$ contained in $\mathcal{S} \times \mathcal{A}$, construct a smooth bump function $\psi$ that equals $1$ on a smaller concentric ball and $0$ outside $B$ (possible since $\mathcal{S}, \mathcal{A}$ open), and define

$$F(s, a) = H(s, a) + \psi(s, a)\Big((y_1 - H(s_1, a_1)) + (J_F - \nabla_s H(s_1, a_1))(s - s_1)\Big), \tag{64}$$

for an arbitrary $C^1$ base map $H$. Then $F(s_1, a_1) = y_1$ and $\nabla_s F(s_1, a_1) = J_F$. Defining $G$ analogously with $J_G$ gives the desired pair; values at $(s_0, a_0)$ can be kept fixed by choosing disjoint supports or applying the same local surgery at $(s_0, a_0)$.

By Jacobian-invariance, $\nabla_{s_1} L_F = \nabla_{s_1} L_G$ at $(s_1, a_0, a_1)$. Subtracting the two chain-rule expressions cancels $\partial \Phi / \partial s_1$ and yields

$$(J_F - J_G)^\top \frac{\partial \Phi}{\partial y_1}(s_1, a_0, a_1, y_0, y_1) = 0. \tag{65}$$

Since $J_F - J_G$ can be any matrix in $\mathbb{R}^{n \times n}$, it follows that

$$\frac{\partial \Phi}{\partial y_1}(s_1, a_0, a_1, y_0, y_1) = 0. \tag{66}$$

Because the arguments were arbitrary, and since $\mathcal{S}$ is connected, we conclude $\partial \Phi / \partial y_1 \equiv 0$, meaning the loss is independent of $y_1$. Therefore there exists a $C^1$ function $\widetilde{\Phi}$ such that for every $F$,

$$L_F(s_1, a_0, a_1) = \widetilde{\Phi}(s_1, a_0, a_1, F(s_0, a_0)). \tag{67}$$

In particular, if two models $F$ and $G$ satisfy $F(s_0, a) = G(s_0, a)$ for all $a \in \mathcal{A}$, then $L_F \equiv L_G$ as functions of $(s_1, a_0, a_1)$, and hence

$$\arg\min L_F = \arg\min L_G. \tag{68}$$

We now construct such a pair $F, G$ but with different feasible sets, contradicting the exact-minimizers assumption. Pick two distinct actions $u, v \in \mathcal{A}$, two distinct states $s_A, s_B \in \mathcal{S}$, and an action $a^\star \in \mathcal{A}$. Using bump-function surgery as above, construct $C^1$ models $F$ and $G$ such that

$$F(s_0, a) = G(s_0, a) \quad \text{for all } a \in \mathcal{A}, \tag{69}$$

and

$$F(s_0, u) = G(s_0, u) = s_A, \tag{70}$$
$$F(s_0, v) = G(s_0, v) = s_B, \tag{71}$$

but with swapped second-step goal reachability:

$$F(s_A, a^\star) = g, \qquad\qquad F(s_B, a^\star) \neq g, \tag{72}$$
$$G(s_A, a^\star) \neq g, \qquad\qquad G(s_B, a^\star) = g. \tag{73}$$

Then $(s_A, u, a^\star) \in \mathcal{M}(F)$ but $(s_A, u, a^\star) \notin \mathcal{M}(G)$, and $(s_B, v, a^\star) \in \mathcal{M}(G)$ but $(s_B, v, a^\star) \notin \mathcal{M}(F)$, so $\mathcal{M}(F) \neq \mathcal{M}(G)$. On the other hand, since $F(s_0, \cdot) = G(s_0, \cdot)$ we have $\arg\min L_F = \arg\min L_G$. By assumption (i),

$$\arg\min L_F = \mathcal{M}(F) \quad \text{and} \quad \arg\min L_G = \mathcal{M}(G), \tag{74}$$

implying $\mathcal{M}(F) = \mathcal{M}(G)$, a contradiction. Therefore, no such $\Phi$ can exist. $\qquad\square$

We introduce the stop-gradient operator $\mathrm{sg}(\cdot)$, where $\mathrm{sg}(x) = x$ during the forward evaluation, but $\nabla \mathrm{sg}(x) = 0$ during the backward pass. The modified objective incorporating the stop-gradient mechanism is:

$$\mathcal{L}_{sg}(\mathbf{s}, \mathbf{a}) = \sum_{t=0}^{T-1} \|s_{t+1} - A\mathrm{sg}(s_t) - Ba_t\|_2^2$$

$$+ \sum_{t=0}^{T-1} \beta_t \|g - A\mathrm{sg}(s_t) - Ba_t\|_2^2 \tag{75}$$

where $\beta_t \geq 0$ are the goal loss coefficients. Note that the target $g$ is effectively applied as a penalty on the state at each step to guide the local optimization.

**Theorem A.6** (Linear convergence to a unique fixed point). *Consider the gradient descent iteration*

$$(\mathbf{s}^{k+1}, \mathbf{a}^{k+1}) = (\mathbf{s}^k, \mathbf{a}^k) - \eta \nabla \mathcal{L}_{sg}(\mathbf{s}^k, \mathbf{a}^k)$$

*where $s_0$ is fixed and gradients are computed with the stop-gradient convention (i.e., treating $sg(s_t)$ as constant during differentiation). Assume the linear dynamics setting and that*

$$\beta_t \in [\beta_{\min}, \beta_{\max}] \quad \text{for all } t, \qquad \beta_{\min} > 0,$$

*and that $B$ has full column rank (equivalently, $\sigma_{\min}(B) > 0$). Then there exists a stepsize $\eta \in (0, \bar{\eta})$, where $\bar{\eta}$ depends only on $\beta_{\min}, \beta_{\max}, \|B\|_2, \sigma_{\min}(B)$ (and not on $T$), such that the induced update operator $\mathcal{T}$ on $\mathbf{z} = (\mathbf{s}, \mathbf{a})$ has a unique fixed point $\mathbf{z}^\star$ and the iterates converge linearly:*

$$\|\mathbf{z}^k - \mathbf{z}^\star\| \leq C q^k \|\mathbf{z}^0 - \mathbf{z}^\star\|, \qquad \text{for some } q \in (0, 1) \text{ and } C > 0.$$

*Proof.* Write the objective (re-indexing the goal term to match the action index) as

$$\mathcal{L}_{sg}(\mathbf{s}, \mathbf{a}) = \sum_{t=0}^{T-1} \|s_{t+1} - A\,sg(s_t) - Ba_t\|_2^2 \ + \ \sum_{t=0}^{T-1} \beta_t \|g - A\,sg(s_t) - Ba_t\|_2^2. \tag{76}$$

Define the residuals

$$r_t := s_{t+1} - As_t - Ba_t, \qquad e_t := g - As_t - Ba_t. \tag{77}$$

Under the stop-gradient convention, $sg(s_t)$ is treated as constant during differentiation, so $s_t$ does not receive gradient contributions through the $A\,sg(s_t)$ terms. It follows that the only state-gradient at time $t$ comes from the appearance of $s_t$ as the *next* state in the previous residual, namely

$$\nabla_{s_t} \mathcal{L}_{sg} = 2r_{t-1}, \qquad t = 1, \ldots, T, \tag{78}$$

with the understanding that $r_{-1} = 0$ if $s_0$ is fixed. Likewise, the action-gradient at time $t$ is

$$\nabla_{a_t} \mathcal{L}_{sg} = -2B^\top r_t - 2\beta_t B^\top e_t, \qquad t = 0, \ldots, T-1. \tag{79}$$

Therefore, gradient descent with stepsize $\eta > 0$ yields the explicit update rules

$$s_t^{k+1} = s_t^k - 2\eta\big(s_t^k - As_{t-1}^k - Ba_{t-1}^k\big), \qquad t = 1, \ldots, T, \tag{80}$$

and

$$a_t^{k+1} = a_t^k + 2\eta B^\top\Big(\big(s_{t+1}^k - As_t^k - Ba_t^k\big) + \beta_t\big(g - As_t^k - Ba_t^k\big)\Big), \qquad t = 0, \ldots, T-1. \tag{81}$$

Stack the variables in the time-ordered vector $\mathbf{z} := (s_1, a_0, s_2, a_1, \ldots, s_T, a_{T-1})$. The updates above define an affine map $\mathbf{z}^{k+1} = \mathcal{T}(\mathbf{z}^k) = J\mathbf{z}^k + c$ whose Jacobian $J$ is block lower-triangular with respect to this ordering: indeed, $(s_{t+1}^{k+1}, a_t^{k+1})$ depends only on $(s_t^k, s_{t+1}^k, a_t^k)$ (and on the fixed constants $g$ and $s_0$) and is independent of any future variables $(s_{t+2}^k, a_{t+1}^k, \ldots)$. Consequently, the eigenvalues of $J$ are exactly the union of the eigenvalues of its diagonal blocks.

To characterize a diagonal block, fix $t \in \{0, \ldots, T-1\}$ and consider the pair $y_t := (s_{t+1}, a_t)$. Conditioned on $s_t$ (which appears only as a constant inside $sg(s_t)$ for differentiation), the update $(s_{t+1}^{k+1}, a_t^{k+1})$ is precisely one gradient step on the quadratic function

$$\phi_t(s_{t+1}, a_t; s_t) = \|s_{t+1} - As_t - Ba_t\|_2^2 + \beta_t \|g - As_t - Ba_t\|_2^2, \tag{82}$$

so the corresponding diagonal block equals $I - \eta H_t$ where $H_t = \nabla_{y_t}^2 \phi_t$ is the constant Hessian with respect to $(s_{t+1}, a_t)$:

$$H_t = 2 \begin{bmatrix} I & -B \\ -B^\top & (1 + \beta_t)B^\top B \end{bmatrix}. \tag{83}$$

Assume $\beta_t \in [\beta_{\min}, \beta_{\max}]$ with $\beta_{\min} > 0$ and that $B$ has full column rank, so $B^\top B \succ 0$. The Schur complement of the $I$ block is

$$(1 + \beta_t)B^\top B - B^\top I^{-1} B = \beta_t B^\top B \succ 0, \tag{84}$$

hence $H_t \succ 0$ for every $t$, with eigenvalues uniformly bounded away from $0$ and $\infty$ as $t$ varies:

$$0 < \mu I \preceq H_t \preceq LI < \infty, \tag{85}$$

for constants $\mu, L$ depending only on $\beta_{\min}, \beta_{\max}, \|B\|_2$, and $\sigma_{\min}(B)$. Choosing any stepsize $\eta$ such that $0 < \eta < 2/L$, we obtain for every $t$ that all eigenvalues of $I - \eta H_t$ lie strictly inside the unit disk, and in particular:

$$\rho(I - \eta H_t) \le \max\{|1 - \eta\mu|, |1 - \eta L|\} =: q < 1, \tag{86}$$

where $q$ is independent of $t$ and $T$. Since $J$ is block lower-triangular and its diagonal blocks are exactly $(I - \eta H_t)$ (up to a fixed permutation corresponding to the stacking order), we conclude

$$\rho(J) = \max_t \rho(I - \eta H_t) \le q_0 < 1. \tag{87}$$

Fix any $q$ such that $q_0 < q < 1$. By applying Gelfand's formula, for any square matrix $M$ and any $q > \rho(M)$, there exists an induced norm $\|\cdot\|_\dagger$ such that $\|M\|_\dagger \le q$. Applying this with $M = J$ yields a norm $\|\cdot\|_\dagger$ satisfying

$$\|J\|_\dagger \le q < 1. \tag{88}$$

Hence, for all $k \ge 0$,

$$\|\mathbf{z}^{k+1} - \mathbf{z}^\star\|_\dagger = \|J(\mathbf{z}^k - \mathbf{z}^\star)\|_\dagger \le \|J\|_\dagger \|\mathbf{z}^k - \mathbf{z}^\star\|_\dagger \le q \|\mathbf{z}^k - \mathbf{z}^\star\|_\dagger, \tag{89}$$

and therefore

$$\|\mathbf{z}^k - \mathbf{z}^\star\|_\dagger \le q^k \|\mathbf{z}^0 - \mathbf{z}^\star\|_\dagger. \tag{90}$$

By norm equivalence in finite dimensions, there exists $C > 0$ such that

$$\|\mathbf{z}^k - \mathbf{z}^\star\|_2 \le Cq^k \|\mathbf{z}^0 - \mathbf{z}^\star\|_2. \tag{91}$$

$\square$

**Notes on the stopgrad optimization.** The optimization indeed converges to a fixed point, but one can show that these stable points in the linear convex case are merely the greedy rollouts towards the goal. Two things make the optimization in our setting nontrivial: the nonconvexity of the world model $F_\theta$, and the stochastic noise on the states $s_t$. We now present some characterization on the distribution of trajectories that our planner tends towards.

Let $F_\theta : \mathcal{S} \times \mathcal{A} \to \mathcal{S}$ be a differentiable world model and define the stop-gradient one-step prediction

$$\mu_t := F_\theta(\bar{s}_t, a_t), \qquad \bar{s}_t = \text{stopgrad}(s_t).$$

Consider the stopgrad lifted objective (cf. Eq. (10))

$$\mathcal{L}(\mathbf{s}, \mathbf{a}) = \sum_{t=0}^{T-1} \|\mu_t - s_{t+1}\|_2^2 + \gamma \sum_{t=0}^{T-1} \|\mu_t - g\|_2^2, \tag{92}$$

and the (no-sync) optimization updates

$$s_{t+1}^{k+1} = s_{t+1}^k - 2\eta_s(s_{t+1}^k - \mu_t^k) + \sigma\,\xi_{t+1}^k, \qquad \xi_{t+1}^k \sim \mathcal{N}(0, I), \tag{93}$$

$$a_t^{k+1} = a_t^k - \eta_a \nabla_{a_t} \mathcal{L}(\mathbf{s}^k, \mathbf{a}^k). \tag{94}$$

Throughout, assume $0 < \eta_s < 1$ (for stability of the state contraction).

**Theorem A.7** (Gaussian tube around one-step predictions). *Fix $\{\bar{s}_t^k\}_{t=0}^{T-1}$ and $\{a_t^k\}_{t=0}^{T-1}$ at iteration $k$, and let $\mu_t^k = F_\theta(\bar{s}_t^k, a_t^k)$. Then the state update* (93) *satisfies the conditional mean recursion*

$$\mathbb{E}\big[s_{t+1}^{k+1} \mid \mu_t^k\big] = (1 - 2\eta_s)\,\mathbb{E}\big[s_{t+1}^k \mid \mu_t^k\big] + 2\eta_s\,\mu_t^k. \tag{95}$$

*Moreover, if $\mu_t^k \equiv \mu_t$ is held fixed, then $s_{t+1}^k$ converges in distribution to a Gaussian "tube" around $\mu_t$:*

$$s_{t+1}^\infty \sim \mathcal{N}(\mu_t, \Sigma_{\text{tube}}), \qquad \Sigma_{\text{tube}} = \frac{\sigma^2}{1 - (1 - 2\eta_s)^2} I = \frac{\sigma^2}{4\eta_s(1 - \eta_s)} I. \tag{96}$$

*Analogously, in continuous optimization-time $\tau$, the limiting SDE*

$$ds_{t+1}(\tau) = -\lambda\big(s_{t+1}(\tau) - \mu_t\big)d\tau + \sigma\, dW_{t+1}(\tau), \quad \lambda > 0,$$

*has stationary law $\mathcal{N}(\mu_t, \frac{\sigma^2}{2\lambda}I)$.*

*Proof.* Rewrite (93) as an affine Gaussian recursion:

$$s_{t+1}^{k+1} = (1 - 2\eta_s)s_{t+1}^k + 2\eta_s\mu_t^k + \sigma\xi_{t+1}^k.$$

Taking conditional expectation yields (95). If $\mu_t^k \equiv \mu_t$ is fixed, the centered process $u^k := s_{t+1}^k - \mu_t$ satisfies $u^{k+1} = (1 - 2\eta_s)u^k + \sigma\xi^k$, i.e. an AR(1) process with contraction factor $|1 - 2\eta_s| < 1$. Its unique stationary covariance $\Sigma_{\text{tube}}$ solves the discrete Lyapunov equation $\Sigma_{\text{tube}} = (1 - 2\eta_s)^2\Sigma_{\text{tube}} + \sigma^2 I$, giving (96). The continuous-time statement is standard, given that $\mu_t$ is fixed. $\square$

**Theorem A.8** (Goal shaping induces goal-directed drift of tube center). *Define $\mu_t^k = F_\theta(\bar{s}_t^k, a_t^k)$ and let*

$$J_t^k := \nabla_{a_t}F_\theta(\bar{s}_t^k, a_t^k) \in \mathbb{R}^{d_s \times d_a}, \qquad P_t^k := J_t^k(J_t^k)^\top \succeq 0.$$

*Assume a first-order linearization in the action step:*

$$\mu_t^{k+1} \approx \mu_t^k + J_t^k(a_t^{k+1} - a_t^k). \tag{97}$$

*Then the action update (94) induced by (92) yields the tube-center evolution*

$$\mu_t^{k+1} \approx \underbrace{\big(I - \alpha\gamma P_t^k\big)\mu_t^k + \alpha\gamma P_t^k g}_{\text{goal-directed averaging (drift)}} + \underbrace{\alpha\, P_t^k\, \varepsilon_{t+1}^k}_{\text{exploration forcing}}, \qquad \alpha := 2\eta_a, \tag{98}$$

*where $\varepsilon_{t+1}^k$ denotes the tube residual $s_{t+1}^k = \mu_t^k + \varepsilon_{t+1}^k$. In particular, if $\mathbb{E}[\varepsilon_{t+1}^k \mid \mu_t^k] = 0$, then*

$$\mathbb{E}[\mu_t^{k+1} \mid \mu_t^k, s_t^k, a_t^k] \approx \big(I - \alpha\gamma P_t^k\big)\mu_t^k + \alpha\gamma P_t^k g, \tag{99}$$

*so in controllable directions the mean prediction $\mu_t$ moves toward $g$ as an averaging step. If $0 < \alpha\gamma\lambda_{\max}(P_t^k) < 1$, this is a contractive averaging step toward $g$ on $\text{Range}(P_t^k)$.*

*Proof.* From (92), only terms at time $t$ depend on $a_t$ via $\mu_t$. Using $\nabla_{a_t}\mu_t = J_t$, the gradient is

$$\nabla_{a_t}\mathcal{L} = 2(J_t)^\top(\mu_t - s_{t+1}) + 2\gamma(J_t)^\top(\mu_t - g). \tag{100}$$

Thus the action step is

$$a_t^{k+1} - a_t^k = -\eta_a\nabla_{a_t}\mathcal{L} = -2\eta_a(J_t^k)^\top(\mu_t^k - s_{t+1}^k) - 2\eta_a\gamma(J_t^k)^\top(\mu_t^k - g).$$

Apply the linearization (97):

$$\mu_t^{k+1} \approx \mu_t^k - 2\eta_a J_t^k(J_t^k)^\top(\mu_t^k - s_{t+1}^k) - 2\eta_a\gamma J_t^k(J_t^k)^\top(\mu_t^k - g)$$
$$= \mu_t^k - 2\eta_a P_t^k(\mu_t^k - (\mu_t^k + \varepsilon_{t+1}^k)) - 2\eta_a\gamma P_t^k(\mu_t^k - g),$$

which simplifies to (98) after substituting $\alpha = 2\eta_a$. Taking conditional expectation and using $\mathbb{E}[\varepsilon_{t+1}^k \mid \mu_t^k] = 0$ yields (99). $\square$

Unlike the stopgrad lifted-state dynamics (Theorem A.7), the rollout distribution is not contracted toward the current rollout, but instead noise accumulates throughout nonlinear iterations of the world model. The stochastic rollout distribution need not concentrate in a local tube around the current deterministic rollout trajectory; it can drift and spread away as the horizon $T$ grows.

**Theorem A.9** (Mean evolution and non-tube behavior of noisy rollouts). *Consider a rollout-based stochastic trajectory generated in model-time:*

$$s_{t+1} = F_\theta(s_t, a_t) + \sigma_{\text{env}}\zeta_t, \qquad \zeta_t \sim \mathcal{N}(0, I), \qquad s_0 \text{ fixed}, \tag{101}$$

*and a dense goal objective along the rollout (e.g. $\sum_{t=0}^{T-1} \|s_{t+1} - g\|^2$). Let $m_t := \mathbb{E}[s_t]$ and $\Sigma_t := \text{Cov}(s_t)$.*

*The rollout mean obeys the exact identity*

$$m_{t+1} = \mathbb{E}[F_\theta(s_t, a_t)]. \tag{102}$$

*In particular, if $F_\theta$ is affine in $s$ (i.e. $F_\theta(s, a) = As + Ba + c$), then*

$$m_{t+1} = F_\theta(m_t, a_t). \tag{103}$$

*For general nonlinear $F_\theta$, a second-order moment expansion yields*

$$m_{t+1} \approx F_\theta(m_t, a_t) + \frac{1}{2}\left(\text{Hess}_s F_\theta(m_t, a_t)\right) : \Sigma_t, \tag{104}$$

*so the mean generally* does not *follow the deterministic rollout $F_\theta(m_t, a_t)$.*

*A first-order linearization gives the approximate covariance propagation*

$$\Sigma_{t+1} \approx G_t \Sigma_t G_t^\top + \sigma_{\text{env}}^2 I, \qquad G_t := \nabla_s F_\theta(\mu_t, a_t). \tag{105}$$

*Proof.* Taking conditional expectation of (101) gives $\mathbb{E}[s_{t+1} \mid s_t] = F_\theta(s_t, a_t)$, and then total expectation implies (102). For affine $F_\theta$, expectation commutes with $F_\theta$, yielding (103). For nonlinear $F_\theta$, expand $F_\theta(s_t, a_t)$ around $\mu_t$ by Taylor's theorem; the first-order term vanishes in expectation and the second-order term produces (104). The covariance recursion (105) follows by linearizing $F_\theta(s_t, a_t) \approx F_\theta(m_t, a_t) + G_t(s_t - \mu_t)$ and computing $\text{Cov}(\cdot)$, adding the independent noise variance $\sigma_{\text{env}}^2 I$. The final "non-tube" claim follows because there is no optimization-time contraction that repeatedly pulls $s_{t+1}$ back toward a moving center (as in (95)); instead the forward propagation (105) typically increases spread and the mean can deviate from the deterministic path by (104). $\square$

Theorem A.7 shows that noisy lifted state updates form an noisy "tube" around the one-step predictions $m_t = F_\theta(\bar{s}_t, a_t)$, keeping exploration local and dynamically consistent in optimization-time. Theorem A.8 then shows that dense one-step goal shaping moves the tube center toward the goal by a preconditioned averaging step, while the dynamics residual contributes approximately zero-mean stochastic forcing that enables exploration without horizon-coupled backpropagation. In contrast, Theorem A.9 shows that noisy rollouts evolve by forward propagation of randomness: the mean follows $m_{t+1} = \mathbb{E}[F_\theta(s_t, a_t)]$ (not generally the deterministic rollout), and the distribution can drift/spread rather than concentrate in a tube around the current plan.

## B. Ablations

To rigorously evaluate the contribution of each individual component within our proposed framework, we conducted an ablation study on the Push-T environment with a horizon of 40 steps. In this experiment, we systematically removed specific modules or mechanisms from the full method while keeping all other hyperparameters constant. This analysis allows us to isolate the impact of important components such as: the noised-states optimization, the GD sync steps, and whether our state-gradient-approach is needed

The results, summarized in Table 3, highlight the critical role of each design choice in achieving robust performance. We observe that the **Full Method** achieves the highest success rate, validating the synergy between the proposed components.

The stopgrad is an important component, without it the state optimization feels "sticky" and is difficult to tune. Noise also remains an important component to allow the planner to explore around local minima.

We also ablate different, simpler options of a noise and gradient-based planner, building off a normal rollout-based GD planner. We ablate with two kinds of noise:

1. *Action noise*, where actions are sampled around Gaussians of a fixed variance $\sigma_a$ before rollout evaluation and gradient computation.

*Table 6.* Hyperparameter sensitivity sweeps for a fixed environment (Push-T, horizon 60). 50 seeds, slightly earlier cutoff (10k steps). Performance is less sensitive to undertuning each parameter, but overtuning can sometimes sharply hinder performance.

| state noise scale | | GD sync interval | | $lr_s$ | | $lr_a$ | |
|---|---|---|---|---|---|---|---|
| Value | Succ. (%) | Value | Succ. (%) | Value | Succ. (%) | Value | Succ. (%) |
| 0 | 22 | 25 | 22 | 0.01 | **24** | 0.0001 | 8 |
| 0.1 | **24** | 50 | **24** | 0.05 | 14 | 0.0005 | 18 |
| 0.5 | 20 | 100 | 20 | 0.1 | 20 | 0.001 | **20** |
| 1.0 | 16 | 200 | 20 | 0.5 | 14 | 0.005 | 10 |
| 2.0 | 2 | 500 | 14 | 1.0 | 6 | 0.01 | 4 |

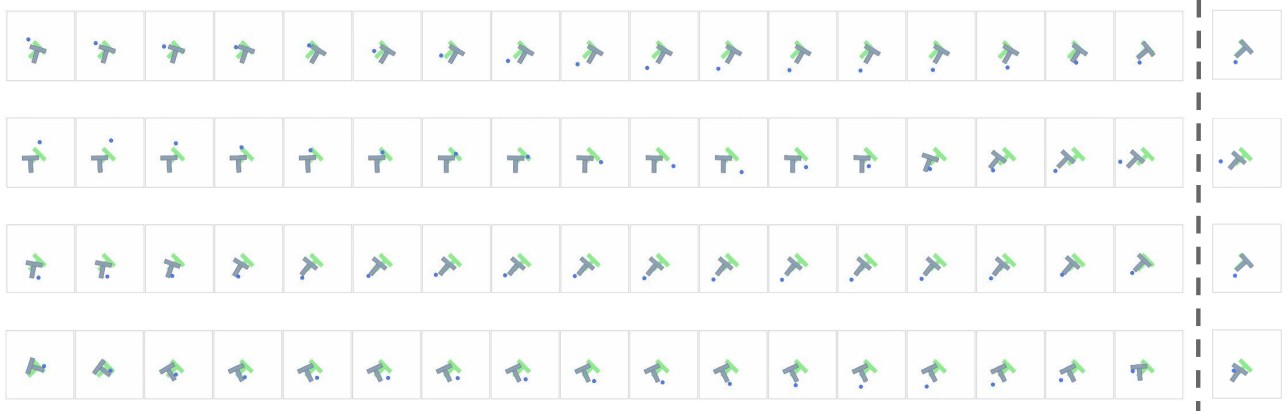

*Figure 6.* Example of converged plans for our planner on the Push-T environment at horizon 80, with goals depicted to the right of the dashed line. The parallel optimization is able to converge more consistently at longer horizons, and find the required non-greedy paths to the goal.

2. *State noise*, where noise is added during the rollout directly to the states:

$$s_{t+1} = F(s_t, a_t) + z, \tag{106}$$
$$z \sim \mathcal{N}(0, \sigma_s I). \tag{107}$$

These ablations are provided in Table 11.

## C. Hyperparameters

The following are hyperparameters used for both the world model and our planner. We provide the model architecture and training hyperparameters used for the DINO-WM model on the PushT environment in Tables 7 and 8. These are the same as in the original paper (Zhou et al., 2024). Table 9 lists the hyperparameters used for our planner.

## D. Additional experiments

| Component | Specification |
|---|---|
| Image size | $224 \times 224$ |
| Frameskip (action chunking) | 5 |
| History length ($T_h$, number of predictions) | 3 |
| Prediction length ($T_p$) | 1 |
| *Encoder (frozen)* | |
| Backbone | DINOv2 (`dinov2_vits14`) |
| Feature key | `x_norm_patchtokens` |
| *Action / Proprio Embedding* | |
| Action dim | 10 (2×5) |
| Proprio dim | 10 (2×5) |
| *Predictor (ViT)* | |
| Depth | 6 |
| Heads | 16 |
| MLP dim | 2048 |
| Dropout | 0.1 |
| Pooling | mean |
| *Decoder (VQ-VAE)* | |
| Channels | 384 |
| Codebook size ($n_{embed}$) | 2048 |
| Residual blocks | 4 |
| Residual channels | 128 |

*Table 7.* DINO-WM architecture for PushT.

| Hyperparameter | Value |
|---|---|
| Epochs | 500 |
| Batch size | 32 |
| Encoder learning rate | $1 \times 10^{-6}$ |
| Predictor learning rate | $5 \times 10^{-4}$ |
| Decoder learning rate | $3 \times 10^{-4}$ |
| Action encoder learning rate | $5 \times 10^{-4}$ |
| Train encoder | False |
| Train predictor | True |
| Train decoder | True |

*Table 8.* Training configuration for DINO-WM on PushT.



*Figure 7.* Example of a bad local minima when relaxing the dynamics constraint into a penalty function. The dynamics are nonzero for the middle transition, but both the states and actions are at a local minima; there's no direction they can move locally to reduce the dynamics loss further.

| Hyperparameter | Value |
|---|---|
| *Optimization* | |
| Optimizer | Adam |
| State learning rate ($\eta_s$) | $10^{-1}$ |
| Action learning rate ($\eta_a$) | $10^{-3}$ |
| Goal loss weight ($\gamma$) | 1 |
| *State Exploration (Langevin-style noise)* | |
| State noise scale ($\sigma_{\text{state}}$) | 0.5 |
| *Periodic GD Refinement* | |
| Use GD sync | True |
| GD interval | 100 |
| GD optimization steps | 25 |
| GD learning rate | 0.1 |
| *Initialization* | |
| State init method | cylinder |
| Initialization noise level | 0.95 |
| Action init distribution | Gaussian |
| Action init mean | 0 |
| Action init std | 0.01 |

*Table 9.* Hyperparameters for GRASP. Cylinder initialization denotes initializing states as a linear interpolation plus Gaussian noise, orthogonalized to the linear interpolation.

*Table 10.* Planning cost (ms/iter) on a single NVIDIA A6000 GPU, measured on Push-T at $H = 12$ (60). Hyperparameters match those used in Table 1. Note that a key benefit of GRASP is parallelizability, which these single-GPU numbers do not fully capture. Numbers may vary depending on GPU size.

| Component | ms/iter |
|---|---|
| **GRASP** (parallel step) | 32.6 |
| **GRASP** (full, average) | 56.8 |
| **GD** (inside GRASP sync) | 158.3 |
| **GD** (standalone) | 705.9 |
| **CEM** (per iteration) | 21,271 |
| **CEM** (per iteration per sample) | 42.5 |

*Table 11.* Stochastic GD Baselines Ablation Study, showing alternatives from our method for a stochastic gradient-based planner. For rollouts of the world model $F_\theta(s, a)$, values of $\sigma_a$ correspond to variance of noise added to actions throughout optimization, and values of $\sigma_s$ correspond to noise added to states directly through the rollout; that is, for rollouts of the world model, $s_{t+1} = F_\theta(s_t, a_i) + \xi, \xi \sim \mathcal{N}(0, I)$. None beat our method's performance, shown in Table 3. Reported time is average time over all experiments, with reported intervals as 95% CI.

| Setting | Accuracy (%) | Time (s) |
|---|---|---|
| *Default Mode (objective.mode=default)* | | |
| $\sigma_s = 0.0$ | $53.0 \pm 4.4$ | $864 \pm 77$ |
| $\sigma_s = 0.5$ | $50.6 \pm 4.4$ | $913.97 \pm 78$ |
| $\sigma_s = 1.0$ | $37.2 \pm 4.3$ | $1149 \pm 75$ |
| $\sigma_a = 0.0$ | $47.6 \pm 4.4$ | $1056 \pm 73$ |
| $\sigma_a = 0.5$ | $53.0 \pm 4.4$ | $864 \pm 77$ |
| $\sigma_a = 1.0$ | $33.4 \pm 4.2$ | $1208 \pm 73$ |
| $\sigma_s = 0.5, \sigma_a = 0.5$ | $53.8 \pm 4.4$ | $854 \pm 78$ |
| *Mode All (objective.mode=all)* | | |
| $\sigma_s = 0.0$ | $54.0 \pm 4.4$ | $865 \pm 77$ |
| $\sigma_s = 0.5$ | $55.8 \pm 4.4$ | $838 \pm 77$ |
| $\sigma_s = 1.0$ | $41.0 \pm 4.4$ | $1111 \pm 75$ |
| $\sigma_a = 0.0$ | $46.6 \pm 4.4$ | $1067 \pm 73$ |
| $\sigma_a = 0.5$ | $52.8 \pm 4.4$ | $873 \pm 77$ |
| $\sigma_a = 1.0$ | $52.4 \pm 4.4$ | $878 \pm 77$ |
| $\sigma_s = 0.5, \sigma_a = 0.5$ | $55.8 \pm 4.4$ | $838 \pm 77$ |

