# OpenReview forum: "Parallel Stochastic Gradient-Based Planning for World Models"
_ICML.cc/2026/Conference — ICML 2026 regular_

### Official Review · Reviewer_fpNS · 2026-02-15

**Soundness:** 3
**Presentation:** 3
**Significance:** 2
**Originality:** 2
**Overall Recommendation:** 4
**Confidence:** 4

**Summary:**

The paper tackles the challenges of sequential planning with world models, which is brittle and hard, especially in large-dimensional vision environments where the states are images. Planning is formulated as the optimization problem of finding the best action sequence that will lead to a desired goal state, and a learned world model is used to imagine the intermediate states leading up to it.

Compared to random shooting, or zero-degree optimization methods, the paper adopts a gradient-based optimization over the optimal actions as well as the intermediate states. This has various difficulties, the biggest of which are optimizing through sequential rollouts of the world model and getting stuck in local minima. To address these, the authors:
- Treat the sequential states as parallel, independent state variables to be optimized;
- Use Langevin-style stochastic optimization in order to decrease the chance of landing in a local, instead of global optimum
- Intermittently perform a standard full-rollout gradient descent step.

Experimental evaluations are performed on the D4RL and DeepMind Control Suite environments.

**Compliance With Llm Reviewing Policy:**

Affirmed.

**Key Questions For Authors:**

Small writing issues to fix
===================

- "due to differentiated through the full rollout" should be "differentiation through the full rollout"
- "brittle or poorly calibrated Jacobians". Jacobian of what with respect to what? Is it the next state with respect to the current actions?
- "This alone would promote trajectories near the starting state, but with a dense one-step goal loss over the full trajectory, converged trajectories for these noisy iterations tend towards the goal." This is very unclear.
- "a deep composition of Fθ can be" should be "a deep composition of Fθ, which can be".
- " does not necessarily have any nice low-dimensional or convex structure". What does "nice" mean here?
- "to find for the solution" needs to be paraphrased
- "learning finding" needs paraphrasing
- "is more intended" needs paraphrasing
- "may ways" should be "many ways"

Questions to authors
============

1. Given that sampling from a diffusion model is very similar to Langevin dynamics, have you considered using diffusion to generate the intermediate states?
2. Please comment on the weaknesses above. Specifically points 1 and 3, and provide reasonable explanations of how your proposed method will or will not address them, or whether these points are not applicable for your method.
3. Could you list state-of-the-art results on D4RL and Deepmind control suite and comment in relation to them?

Finally, an interesting and relevant planning paper is https://arxiv.org/abs/2511.11043
It performs "multiple shooting" like MPPI but also does a step of gradient descent over the actions.

Overall, it's a good paper. I would like to see some commentary on my conceptual weaknesses raised above. Also, if possible, more baselines, like a fully-reactive policy rolled out at test time.

**Limitations:**

yes

**Strengths And Weaknesses:**

Overall, it has all the elements of a good contribution - it's well-designed, clearly motivated and well-written. I like the approach and the direction. That being said, I'm not sure the proposed contributions are enough to unlock the general, highly-desired, planning capabilities in more difficult and realistic settings. Likely, the proposed method is part of the solution, but not the whole solution.


Strengths
=======
1. Problem formulation is clear and standard - test-time planning using the world model. Similar ideas have been explored extensively in model-based RL and in that sense, the paper is well-grounded and the problem it tackles is very important to the community. Making a good contribution to this problem can have a wide ripples in terms of impact also in other areas of ML.
2. Generally, very well written and clear. Equations, text, figures and their captions are all very understandable. There are occasional sentences that need paraphrasing but overall this doesn't hurt the content and is not something that cannot be fixed in the camera ready.
3. The proposed technical contributions -- parallel states, Langevin dynamics, and full-rollout synchronization -- are well-designed and motivated from first principles.


Weaknesses
=========
In general, test-time sequential action optimization using a world model is not an easy problem, and unfortunately, I don't think the authors' proposed design of the optimization approach is a long-term solution that can be applied to larger and more complicated settings. That is, the contributions can certainly be useful, but are they enough? Consider the following:
1. To optimize the intermediate states in a parallel way, as opposed to a sequential one, you need to **know the length** $T$. While this assumption is standard in trajectory optimization, it may limit applicability in tasks where the required plan length is unknown or must be determined adaptively. It would be helpful for the authors to discuss whether the method could be extended to variable-horizon or receding-horizon settings.
2. The other problem with optimizing intermediate states is that **direct gradient over a learned world model fails to keep the states "physically consistent"**. What I mean is that in many environments the state sequence is generated through actual physical laws, and when you use a learned world model, without incorporating these physical laws (themselves unknown), the predictions end up being physically impossible. This is very clearly visible in driving for example. Real trajectories are smooth, whereas generated trajectories, even if they roughly seem plausible, often violate certain acceleration/jerk thresholds of the vehicle dynamics. And this all happens because the world model predicts the next state without knowing or incorporating the underlying dynamics, oftentimes the underlying analytic equations.
3. Eqn. 9 is very tricky because it encourages the agent to reach the goal state **as fast as possible**. For some environments, like a vehicle trying to reach a desired location, but waiting at a red light, the proposed loss would be detrimental, because it will encourage the agent to drive the red light and reach the goal as fast as possible. It is not at all obvious that every single transition should bring you closer to the goal state, which is what this loss encourages. Or perhaps these are latent states here, which can contain implicitly the notion of time in them? Please comment on this.

In addition to the above conceptual weaknesses, I find the experiments slightly underwhelming, because these environments are relatively old by now. There are plenty of robotics environments which are good testing grounds for such planning algorithms. In addition, there are no non-planning baselines, so it's hard to judge the performance of the proposed method compared to more mainstream solvers and reactive policies. Please consider comparing against state-of-the-art methods on these environments.

---

> ### Author Rebuttal · Authors · 2026-03-31
>
> Thank you for your thoughtful review and for highlighting the potential impact and clarity of the paper. We appreciate your careful feedback. Below, we respond to the main weaknesses and questions you raised.
>
> ---
>
> **W1**: One way to incorporate into a variable or receding horizon closed-loop planner is the direct way: replace the fixed-horizon open-loop sub-solver in MPC with GRASP, and proceed like this. But another we find more interesting and more suited for a planner like GRASP is a hierarchical planner (e.g. [[1](https://arxiv.org/pdf/2406.11506)]): one normal receding-horizon MPC at a higher frequency, and a longer-horizon planner in parallel at a lower frequency (which would be GRASP in this context), and in these settings it is flexible to choose a horizon that’s simply high enough for the given problem while the lower-level MPC level remains adaptive.
> Our goal for this paper though is to isolated the open-loop planning capabilities as it is a critical component for closed-loop and variable-horizon systems, and leave such studies for future work.
>
> **W2**: While collocation-based methods may directly exploit these inconsistencies more, GRASP was developed to limit this exploitation as much as possible by limiting dependency on the state gradients, while still keeping the long-horizon benefits and parallelism of collocation-based planners.
> However, we do not believe this issue is unique to our planner. More broadly, it arises whenever planning is performed with a learned world model whose rollouts are not fully faithful to the underlying dynamics. Training a world model with more physically consistent dynamics is an interesting direction of work, but outside the scope of our paper.
>
> **W3**: We agree that, in isolation, such a goal-reaching loss would be problematic. However, the dense goal loss is not in isolation, and is balanced both by the stopgrad-modified dynamics loss (where states are biased more towards the current trajectory irrespective of the goal) and the occasional GD sync steps. Looking at both Table 2 from the paper (sync steps ablation) and the following ablation on how many stochastic Eqn. 10 steps to do between GD phases:
>
> | Parameter  | Sweep values   | Success rates  |
> | - | - | -|
> | `gd_interval`    | {0 (GD), 50, 200, 500, no sync}     | {16.4%, 21.2%, 24.2%, 5.6%, 0.0%}  |
>
> We can conclude that both components of the planner are important (notably at the optimum, there are more Eqn. 10 steps (200) than GD steps (25)). And indeed, the full GRASP planner still finds many non-greedy plans in the Push-T environment. Please also see our rebuttal to AZjb (**Q1**) for more context on this particular loss.
>
> **New experiments**: Please see our rebuttal to o5kt in their **W3**.
>
> **Q1**: This is an interesting direction. In this work, however, our goal was to study the more general planning problem and develop a planner that can be applied to a broad class of learned world models, rather than relying on a particular generative model class.
>
> That said, we agree that diffusion-based models (either diffusion world models or pure-state action-free diffusion models) appear to be a natural fit for collocation-style planning, either using the learned latent score function as a smoothed prior for the dynamics loss or as a projector for states during optimization, respectively. We view this as a promising specification/extension rather than something required by our method.
>
> **Q2**: Weakness responses.
>
> **Q3**: The difficulty for our setting is purely image-based world models (not over intrinsic features or using compressed latent spaces, e.g. in [[1](https://arxiv.org/abs/2106.13229)]), in order to evaluate our planner in the context of a more general world model with a bigger latent space. For a recent paper comparison in a related setup, the reported long-horizon Push-T results in e.g. [[2](https://arxiv.org/pdf/2512.09929)] report at H=50 that DINO-WM in a closed-loop setting (replanned 20 iterations through plan execution) achieves 16% success for the base DINO-WM and 26% for their adversarial WM, while GRASP achieves 43.4% in the same setting on base DINO-WM but rolled out open-loop from the origin.
>
> **Relevant paper**: Indeed an interesting and relevant paper, we can add a citation for the camera-ready. This appears to fall under the category of hybrid planners (e.g. CEM-GD [[1](https://arxiv.org/abs/2112.07746)]), and as described in the conclusion we find this to be an interesting future direction but leave it for future work, and focus on the individual planners’ performance for this paper.
>
> We are also happy to consider providing a closed-loop baseline for the camera-ready to illustrate further that GRASP can be adapted into a closed-loop setting.
>
> ---
>
> If we have addressed your questions/concerns, we would greatly appreciate your reconsideration of the score. If any questions remain, we would be very happy to clarify them. Thank you again for your time and thoughtful review.

---

> > ### Author Rebuttal · Reviewer_fpNS · 2026-04-03
> >
> > Thank you for the rebuttal and the clarifications. I find the comments over points W1, W2, and W3 informative and sufficiently realistic in their judgement. I think the important thing is to acknowledge them, in order to better understand the limitations of the proposed approach and how it fits in the broader picture of test-time planning.
> >
> > One thing I would mention is that it *may* not always be the best strategy to study individual components (e.g. planners) in isolation. Sometimes it is the combination of the method and the task setting that together contain the most insights. That is why my point was to think about the proposed stochastic parallel planning method in the context of the problem that we want to solve.
> >
> > Overall, it's a good paper. I'll keep my weak accept score.

---

> > > ### Author Response · Authors · 2026-04-05
> > >
> > > Thank you for your thoughtful engagement throughout the review and for acknowledging that your concerns have been adequately addressed from the rebuttal. We appreciate the substantive commentary in the initial review and follow-up.
> > >
> > > We agree that studying GRASP's closed-loop interactions with physical world properties (e.g. feedback and structured noise) is an important direction for future work. For this paper, however, our focus is on the optimization properties of world model planners as they relate to the deep learning-based world model (e.g. isolating the adversarial robustness issue depicted in Fig. 3), and thus isolate the study to these properties in particular.
> > >
> > > We will incorporate the discussion here into the camera-ready version. If there are any remaining questions/concerns keeping you from raising your score, we are happy to address them in the time remaining.
> > >
> > > Thank you again for your careful review.

---

### Official Review · Reviewer_o5kt · 2026-03-12

**Soundness:** 3
**Presentation:** 3
**Significance:** 2
**Originality:** 3
**Overall Recommendation:** 4
**Confidence:** 3

**Summary:**

Summary:
The authors propose a gradient-based method for performing long-horizon planning using learned vision-based world models. The proposed method performs parallelized lifted planning, where dynamics consistency is enforced through a penalty function. This makes the optimization fast. To avoid local optima, a common problem in gradient-based planning, the authors propose adding noise to the states. The planning procedure is made more robust by only using gradients on the action inputs, and the authors present a theoretical argument for their design choice. In the experiments, the method outperforms baselines on long-horizon open-loop planning.

**Compliance With Llm Reviewing Policy:**

Affirmed.

**Final Justification:**

I think the rebuttal clarified the main concern I had about the evaluation procedure (W1). With that clarified, I think the paper exceeds the bar for publication at ICML.

**Key Questions For Authors:**

- See weaknesses above.

**Limitations:**

Yes

**Strengths And Weaknesses:**

Strengths:
- The paper has interesting and promising ideas and their combinations, such as the stop-gradient with respect to state inputs, and the exploration through the Langevin-style update on the states only, which are also well-motivated from a theoretical and practical angle. The theory gives useful intuition
- The examples in Figure 3 are illustrative for understanding why the stop-gradient on states matter.
- The performance of the proposed method on long-horizon open-loop visual planning tasks with a learned world model is clearly superior to that of the baselines.
- The wall-clock speed of the planner is excellent, especially at long horizons.

Weaknesses:
- The evaluation is done in the learned world model, not in the actual environment. In-model evaluation cannot necessarily detect model exploitation. Did I somehow misunderstand the paper?
- The theory feels disconnected from the actual algorithm that works, because the theory analyzes the case without the periodic sync, which according to the ablation study is the most critical part of the algorithm (1.6% accuracy without sync, Table 2).
- The planner is not a general planner, rather, it's specifically for goal-reaching. The model is evaluated in an environment where the underlying state is low-dimensional. The action spaces are also low-dimensional (2D). It's hard to say, to what extent the findings in the ablation generalize, because most ablations are done on one environment. The method is also only evaluated with a single world model (DinoWM).
- Long-horizon open-loop planning is arguably a somewhat specialized problem. It can be important for zero-shot generalization, but in most problem settings, you would typically learn a value function which would enable reducing the planning horizon. If no value function were learned, hierarchical planning to reduce the effective planning horizon would be a promising alternative. The paper does not discuss or compare against such approaches. In short-horizon planning problems, this method did not show significant gains over baselines.

---

> ### Author Rebuttal · Authors · 2026-03-31
>
> Thank you for your thoughtful review and for highlighting the combination of ideas and speed of the proposed planner. We appreciate your careful feedback. Below, we respond to the main weaknesses and questions you raised.
>
> ---
>
> **W1**: We apologize if this was unclear. The planner optimizes action sequences using the learned world model, but *all reported success rates are computed by executing the planned actions in the actual environment*. We will make sure this is clear in the camera-ready.
>
> **W2**: The intermediate steps act as stochastic guidance towards fully solved plans, and the theory provided in the appendix is to characterize the type of exploration done in the intermediate steps, as this is the main novelty of GRASP. Since the reshaped loss’s minima don’t precisely correspond to goal-reaching plans, some sync step (like a GD sync step) is needed. However, Theorem 1 from our paper shows this is actually *necessary*: you cannot have a planning optimization objective that is both (a) invariant to state gradients (which can be adversarial, as depicted in Figure 3), and (b) doesn’t need any help from something like GD sync steps. Since we get better success rates by adding more GRASP exploration steps:
>
> | Parameter           | Sweep values                       | Success rates       |
> | - | -- | - |
> | `gd_interval`       | {0 (GD), 50, 200, 500, no sync}     | {16.4%, 21.2%, 24.2%, 5.6%, 0.0%}  |
>
> We can conclude that both components of GRASP (the parallel stochastic detached part and the GD sync steps) are necessary. Note further the optimal success rate falls when GRASP steps overtake GD steps (25 iteration per GD sync phase).
>
> **W3**: We have now evaluated our planner on two different world models: [jepa-wm](https://arxiv.org/abs/2512.24497) and [le-wm](https://arxiv.org/abs/2603.19312). We get good results on a further collection of tasks:
>
> | Environment     | Horizon (environment steps) | Num trials | CEM |   GD |   GRASP  |
> | - | - | - | - | - | - |
> | Reacher (le-wm)        |      15 (75) |       50 | 38% |      42% |           **70%** |
> | PushT (le-wm) | 15 (75) |50| 22% | 26% | **46%*** |
> |Cube (le-wm) | 10 (50) |50| 58% | 56% | **66%**** |
> | Metaworld Reach (jepa-wm) |       6 (30) |       16 | 12% | 6% |       **19%****|
>
> We found the following small modifications helpful in some environments:
> - *: replacing the GD sync with a small CEM sync
> - **: linearly decaying the state noise and goal loss coefficient (gamma in Eq. 10) between each GD sync step
>
>  The le-wm results come from very similar settings from the paper (of which hyperparameters stayed constant throughout all our original paper’s experiments), just tuning learning rates. We hope this provides more evidence that GRASP extends to other world models. We will run on more seeds for the camera-ready.
> More general end-state loss functions (or dense loss functions) can also be naturally considered for GRASP by optimizing the goal state $g$ directly with respect to the reward $r(g)$. We focus on goal-reaching for simplicity of the presentation of our new planner, but more general rewards is an interesting direction for future work.
>
> **W4**: We do not view GRASP (or other world model planners) as replacements for value functions or hierarchical planning, but rather complementary in pipelines that benefit from direct world model planning to provide target trajectories (e.g. [[1](https://arxiv.org/pdf/2511.00423)]). For our paper, we focus on the open-loop long-horizon setting because this remains an important subproblem for zero-shot generalization, offline settings, and as a component that could ultimately be combined with learned values or hierarchical structure. Our goal here is therefore to isolate the open-loop planning capabilities of the planner itself; extensions that combine GRASP with value functions or hierarchical planning are promising, but beyond the scope of the current paper.
>
> ---
>
> If we have addressed your questions/concerns, we would greatly appreciate your reconsideration of the score. If any questions remain, we would be very happy to clarify them. Thank you again for your time and thoughtful review.

---

> > ### Author Rebuttal · Reviewer_o5kt · 2026-04-04
> >
> > Thank you for clarifying W1 and for presenting results on two additional world models. I still view the theory/practice gap as a minor weakness, with the support for the method being mostly empirical. I find the evaluation and the scope somewhat narrow. But after the rebuttal, I think the paper is above the bar

---

> > > ### Author Response · Authors · 2026-04-05
> > >
> > > Thank you for raising your score, your compliment of the paper, and for the constructive engagement throughout the review process. We are glad the additional experiments and clarifications were helpful.
> > >
> > > Are there any remaining questions on the theoretical portion of the paper? Appendix A.1 provides optimization conditioning motivation for studying collocation overall, A.2 establishes regularity benefits for our state-noised descent, Theorem 1 (proof in A.4) establishes the necessity for a GD-like sync step (no objective can be state-gradient-insensitive and preserve the set of strict minimizers), and Theorems 8 and 9 in Appendix A.4 provide distributional characterizations in the convex case for what GRASP steps gravitate towards vs. noisy serial rollout methods.
> > >
> > > If there is any particular connection you would like to see strengthened, or if there is anything further we can clarify, we are happy to do so.
> > >
> > > Thank you again for your time and careful review.

---

### Official Review · Reviewer_X2VG · 2026-03-12

**Soundness:** 3
**Presentation:** 3
**Significance:** 3
**Originality:** 3
**Overall Recommendation:** 4
**Confidence:** 3

**Summary:**

The paper introduces a new gradient-based planning approach for world models, focusing on overcoming common challenges like local minima and optimization instability over long horizons and high-dimensional state spaces. The method involves a "lifted" planning approach, where intermediate states are optimized in parallel, avoiding the need for serial rollouts. Additionally, the method incorporates stochastic updates to promote exploration, particularly useful in escaping local minima. It offers a robust planner that performs well in both short-term and long-term planning tasks in various visual control environments.

**Compliance With Llm Reviewing Policy:**

Affirmed.

**Key Questions For Authors:**

In what cases can your planner outperform existing methods in short-horizon tasks? It would be interesting to see if there are specific task types or domains where the lifted, parallelized approach can provide additional benefits over simple methods like CEM or GD.

How sensitive is the planner to model inaccuracies or real-world environmental noise? Given that the approach relies on learned world models, further analysis of robustness against such imperfections would provide insight into its practical deployment.

Could hybrid planning approaches combining your method with CEM or other baseline methods improve the overall performance? Have you considered hybridizing the stochastic gradient-based planning with other sampling-based methods to cover scenarios where one method excels over the other?

**Limitations:**

The authors adequately discuss the limitations of their work, particularly in terms of its performance in short-horizon tasks and the complexity of hyperparameter tuning. However, further exploration into the impact of model inaccuracies and real-world uncertainties would strengthen the paper. Additionally, the potential societal implications of implementing this approach in real-world systems could be more explicitly addressed, especially in safety-critical applications like robotics or autonomous vehicles.

**Strengths And Weaknesses:**

Strengths:

Originality and Contribution:

The paper presents a novel approach to parallelizing gradient-based planning by treating intermediate states as independent optimization variables. This avoids serial rollouts, thus improving computational efficiency.

It introduces stochasticity to aid in exploration and escape from local minima, making it particularly useful for long-horizon planning.

The method avoids issues with brittle gradients through the learned world model by only differentiating over actions, not the states.

Performance:

The proposed planner outperforms traditional methods like Cross-Entropy Method (CEM) and vanilla Gradient Descent (GD) in long-horizon tasks, both in success rate and time to convergence. The results in Figure 5 and Table 1 provide concrete evidence of its efficiency.

The paper includes a comprehensive ablation study (Table 2) demonstrating the effectiveness of various components (state gradient detaching, GD sync steps, noise levels) in enhancing performance.

Clarity of Approach:

The methodology is well-articulated with visual aids (Figures 2, 3, 4) that clearly show how the proposed approach decouples state dynamics for parallel optimization.

The integration of stochastic updates and gradient-based planning with practical applications to visual control environments is explained clearly and with appropriate figures for illustration.

Weaknesses:

Limited Impact in Short Horizons:

While the approach is robust in long-horizon planning, its performance in short-horizon tasks does not outperform existing methods like CEM or GD (as shown in Table 3 and Table 4). This limits the broader applicability of the method, especially in scenarios where simpler methods are already sufficient.

Complexity and Hyperparameter Sensitivity:

The planner relies on several hyperparameters such as noise level, gradient detachment, and synchronization frequency. These may require careful tuning, especially for non-expert users.

The paper acknowledges the added complexity introduced by these parameters but does not provide sufficient guidance on how to handle this tuning in practice.

Scalability to High-dimensional Systems:

The method addresses some issues with high-dimensional state spaces, but further discussion is needed on how it scales to even higher-dimensional or more complex systems that might involve more intricate world models.

In some cases, the world model’s Jacobian can still be poorly conditioned, and further strategies for improving stability in such high-dimensional systems could be beneficial.

Lack of Robustness Against Real-World Uncertainties:

While the paper addresses adversarial robustness in the state gradients, it does not explore real-world uncertainties (e.g., sensor noise, imperfect world model accuracy) that might affect the planner’s performance in practical applications.

---

> ### Author Rebuttal · Authors · 2026-03-31
>
> Thank you for your thoughtful review and for highlighting the novelty and thorough ablations. We appreciate your careful feedback. Below, we respond to the main weaknesses and questions you raised.
>
> ---
>
> **Short horizon**: In short-horizon settings, simpler baselines such as CEM or GD are often already sufficient, so we do not expect large gains there. Our claim is instead that GRASP remains competitive at short horizons while providing its clearest benefit as the horizon increases, which is demonstrated in the results; GRASP has comparable results to all baselines (e.g. confidence interval for 500 trials is ~2.6% at success 0.9), and is either better or within a 95% CI of GD in success rate.
>
> ** Hyperparam sensitivity**: We provide some self-ablations in Table 2 and Table 5 to demonstrate the benefit of each component and the overall noising strategy. As far as individual parameter sensitivity, we find most parameters are relatively easy to tune, with the main sensitive parameter being the action learning rate. *We also provide more experiments varying hyperparameters* (please see our rebuttal to AZjb under **W4**).
>
> **Scalability to high-dim systems**: While analysis on sharper non-greedy plans is an interesting direction for future work, we note that there are core theoretical ideas supporting GRASP’s scalability over other commonly used planners like CEM or GD, such as: (1) better scaling to longer horizons than GD as demonstrated in Appendix A.1 and empirical evidence that gradient-based methods scale better with action dimension than CEM [[1](https://arxiv.org/pdf/2004.08763)], and (2) better scaling with sharper gradient landscapes via the gradcut design for GRASP.
> We also provide further results on new world models (le-wm and jepa-wm), which show favorable results for GRASP (please see our rebuttal to o5kt under their **W3**).
>
> **Noise robustness**: Intuitively, small i.i.d. Gaussian state noise to the WM outputs should not affect performance much, as we see the state noise parameter is relatively insensitive with respect to planning success:
>
> | Parameter     | Sweep values     | Success rates         |
> | -- | --| --- |
> | `state_noise_scale` | {0.0, 0.1, 0.25, 0.5, 1.0, 2.0}      | {20.0%, 20.0%, 10.0%, 10%, 10.0%, 0.0%} |
>
> For more general structured noise, it’s harder to make any rigorous claims. We agree this is important, but it is somewhat orthogonal to the main contribution of the paper. Our focus is on planner stability given a learned world model; if the world model itself is inaccurate under sensor noise or distribution shift, any planner built on top of it will be limited by that error. We further expect that the robustness of world models will get better with current and future works, e.g. [[2](https://arxiv.org/abs/2512.09929)].
>
> **Q1**: For higher action dimensions, we expect gradient methods to outperform CEM-based methods [[1](https://arxiv.org/pdf/2004.08763)], and we do have some preliminary results now on a higher-dimensional action space (4 and 5 for metaworld-reach and cube respectively) and a shorter horizon (H=6 WM steps / 30 env steps):
> | Environment     | Horizon (environment steps) | Num trials | CEM |   GD |   GRASP  |
> | --------------- | ------: | -------: | --: | -------------------: | ------- |
> | Metaworld Reach ([jepa-wm](https://arxiv.org/abs/2512.24497)) |       6 (30) |       16 | 12% | 6% |       **19%***|
> |Cube ([le-wm](https://le-wm.github.io/)) | 10 (50) |50| 58% | 56% | **66%**** |
> (See our rebuttal to o5kt for details on (*) and (**))
>
> **Q2**: We have some evidence and reason for relatively small Gaussian noise to be permissible (see response to **noise robustness**), but for the general structured noise case we view this as an important but orthogonal line of work.
>
> **Q3**: We do expect this to improve performance and consider it an interesting line for future work (e.g. [CEM-GD](https://arxiv.org/abs/2112.07746) style). For this paper, we wanted to isolate the individual planners to investigate and evaluate them individually.
>
> ---
>
> If we have addressed your questions/concerns, we would greatly appreciate your reconsideration of the score. If any questions remain, we would be very happy to clarify them. Thank you again for your time and thoughtful review.

---

> > ### Author Rebuttal · Reviewer_X2VG · 2026-04-03
> >
> > Thank you for the reply. I will keep my score.

---

> > > ### Author Response · Authors · 2026-04-03
> > >
> > > Thank you for your rebuttal response. If your concerns have been adequately addressed, we ask kindly for you to update your score if all weaknesses have been addressed. If there is anything further we can address, please let us know.
> > >
> > > Thank you again.

---

### Official Review · Reviewer_AZjb · 2026-03-13

**Soundness:** 3
**Presentation:** 3
**Significance:** 3
**Originality:** 3
**Overall Recommendation:** 4
**Confidence:** 4

**Summary:**

This paper proposes a parallel stochastic gradient-based planner for vision-based world models. The authors address the challenges of long-horizon planning and gradient instability in high-dimensional latent spaces by introducing "virtual states" as independent optimization variables. This method enforces dynamics through soft constraints in the objective function. This allows the optimization to be parallelized across the entire planning horizon. Additionally, the framework incorporates stochasticity into the state variables to aid exploration and modify the gradient structure to descend towards valid plans while only requiring action-input pairs to mitigate sensitive gradients inherent in complex visual world models. The method is evaluated on the PointMaze, Wall-Single, and PushT benchmark, demonstrating an ability to find non-greedy paths in long-horizon tasks.

**Compliance With Llm Reviewing Policy:**

Affirmed.

**Final Justification:**

As the authors have adequately addressed my questions, I'd like to maintain my positive score.

**Key Questions For Authors:**

* Q1. How critical is the periodic full-rollout synchronization step in preventing the optimizer from settling on dynamics-inconsistent virtual-state plans? Do you have quantitative diagnostics on the residual inconsistency before and after sync?

* Q2. Can you provide a cleaner compute breakdown versus the sequential GD and CEM baselines, separating optimization-iteration count from per-iteration cost?

* Q3. In obstacle-rich settings that require precise non-greedy intermediate states, how reliable is the stochastic exploration compared with derivative-free methods such as CEM?

**Limitations:**

yes, in Section 6

**Strengths And Weaknesses:**

**Strengths:**

* The paper is well-written and structured. Moving from sequential shooting to a collocation-like approach using virtual states is a theoretically sound way to improve the optimization landscape for long-horizon planning. It breaks the long-chain dependencies that make standard backpropagation through time (BPTT) difficult.

* The paper provides a simplified theoretical analysis explaining why the lifted formulation is better conditioned than shooting and how state noise smooths the optimization landscape.

* Planning over an 80-step horizon directly from pixels is non-trivial and the ability to converge consistently on tasks like Push-T suggests the method can be robust for complex spatial reasoning.

* By decoupling the states, the planner exposes substantial parallelism across the horizon, which is an important computational advantage over serial rollout-based optimization.

**Weaknesses:**

* W1. The rely on "soft dynamics constraints" is a potential point of failure. If the dynamics penalty is not weighted heavily enough, the optimizer might find "ghost" trajectories, solutions that look optimal but are physically impossible because they violate the underlying world model's dynamics. The paper proposes several stabilizers, but the interaction between the dynamics penalty, goal-shaping weight, noise scale, and sync frequency is not analyzed in much depth.

* W2. While applying this to latent world models is a fresh take, the core concept, treating states as optimization variables and using soft constraints to handle dynamics, strongly resembles classical direct collocation or ADMM-based trajectory optimization methods. The paper would be more convincing if it explicitly compared its performance and convergence properties to these well-established optimization techniques.

* W3. The primary results are centered on the simulated environments. While Push-T is a respectable benchmark for long-horizon manipulation, it is relatively slow-moving and quasi-static. It remains unclear how well the method would transfer to faster or more contact-rich systems, where learned world-model gradients may be even less reliable.

* W4. The method introduces several new tuning knobs, such as the dynamics penalty weight and the Gaussian smoothing parameters. The authors do not provide a clear sensitivity analysis, leaving me to wonder how much manual effort is required to port this to a new environment.

---

> ### Author Rebuttal · Authors · 2026-03-31
>
> Thank you for your thoughtful review and for highlighting our long-horizon pixel-space results. We appreciate your careful feedback. Below, we respond to the main weaknesses and questions you raised.
>
> ---
>
> **W1**: We appreciate this concern, but we respectfully disagree that this issue is not analyzed in depth. The paper already includes ablations on the key mechanisms introduced; Table 2 provides the primary method ablations, where we show performance using various amounts of sync steps, noise levels, and the detached gradient method (not this also ablates using the goal loss or not), and Table 5 showing further ablations over different noising strategies (both parallelized and serialized strategies), to show our particular way of noised-based exploration effectively finds correct trajectories at longer horizons. For other parameter sensitivities, please refer to the new results discussed in W4.
>
> **W2**: Our method is indeed inspired by classical state-optimization methods such as direct collocation, multiple shooting, and ADMM-style trajectory optimization, as we mention in the introduction and related work (e.g. [[1](https://www.frontiersin.org/journals/robotics-and-ai/articles/10.3389/frobt.2022.890385/full)[2](https://ieeexplore.ieee.org/document/9888973)[3](https://arxiv.org/abs/2106.13229)). We do not claim novelty in the abstract optimization template itself, but in making this style of optimization work reliably for modern learned visual world models with brittle state gradients: the stop-gradient design, stochastic state updates, and periodic synchronization together stabilize optimization without requiring any modification to world-model pretraining. [Latco](https://arxiv.org/abs/2106.13229) also acts as a collocation baseline (which also uses more nuanced classical optimizers), and we find that purely classical forms of collocation-based planning are not enough to scale up to modern world models.
>
> **W3**: We agree this is an important question, and *we now include additional results beyond Push-T on more robotic and visually complex environments* (please see our rebuttal to o5kt under their W3).
>
> **W4**: We actually use the same set of hyperparameters for all experiments in the original submission, we realize the paper’s current wording is slightly misleading and will fix for the camera-ready. For a fixed longer-horizon environment (here we chose push-t at H=60), here are some more sweeps over parameters:
>
> | Parameter   | n_trials | Sweep values | Success rates  |
> |-|-|-|-|
> | `state_noise_scale` | 10 |0.0, 0.1, 0.25, 0.5, 1.0, 2.0      | 20.0%, 20.0%, 10.0%, 10%, 10.0%, 0.0% |
> | `lr_s`    | 10 |  0.01, 0.05, 0.1, 0.5, 1.0    | 20.0%, 10.0%, 10%, 20.0%, 0.0%    |
> | `gd_interval`  | 36 | 0 (GD), 50, 200, 500, no sync     | 16.4%, 21.2%, 24.2%, 5.6%, 0.0%  |
> | `lr_a`   | 36  | 0.0001, 0.0005, 0.001, 0.005, 0.01 | 11.1%, 22.2%, 13.2%, 0.0%, 0.0%   |
>
> `lr_a` is the most sensitive, but good performance still resides within an order of magnitude. Parameters were also relatively straightforward to tune to some new environments/wm’s like le-wm Reacher (lr_a -> 0.005). We will run over more seeds for the camera-ready.
>
> **Q1**: It is critical for complete success, since the minimizers for our reshaped parallel objective don’t precisely match with goal-reaching plans. However, Theorem 1 from our paper shows this is actually *necessary*: you cannot have a planning optimization objective that is both (a) invariant to state gradients (which can be adversarial, as depicted in Figure 3), and (b) doesn’t need any help from something like GD sync steps. The gd_interval sweep steps above show that the internal parallel steps help with guidance towards correct paths, as the optimal success rate falls when GRASP steps overtake GD steps (25 iteration per GD sync phase).
>
> **Q2**: While there are some important subtleties for cost (one main benefit of GRASP is parallelizability), here are some ms/iter stats on our current hardware (single GPU, also evaluated on Push-T at H=12 (60)):
>
> | Planner | ms/iter | Relative ms/iter |
> | - | - | - |
> | GRASP   |   109.9 |   1.0× |
> | GD   |   705.9 |    6.4× |
> | CEM    |    23753.2 |    216× |
>
> **Q3**: The stochastic component is intended to address settings that require non-greedy intermediate states, and the PushT experiments together with the ablations against pure GD suggest that this guided exploration helps the optimizer escape greedy local minima. That said, we have not rigorously isolated the setting of very sharp or precise non-greedy solutions in a dedicated experiment. We expect that such cases may require more careful tuning of the noise schedule or a stronger exploration mechanism, which we view as an interesting direction for future work.
>
> ---
>
> If we have addressed your questions/concerns, we would greatly appreciate your reconsideration of the score. If any questions remain, we would be very happy to clarify them. Thank you again for your time and thoughtful review.

---

> > ### Author Rebuttal · Reviewer_AZjb · 2026-04-04
> >
> > Thank you for the detailed response. I'm curious about:
> >
> > 1. Are there existing non-wm baselines for the tasks in your experiments (e.g. on/off-policy RL)? How did they perform?
> >
> > 2. Can you learn a single world model for these tasks?

---

> > > ### Author Response · Authors · 2026-04-05
> > >
> > > Thank you for your response and for thoughtful follow-up questions, and we are happy to hear your review concerns have been adequately addressed. We are happy to address both.
> > >
> > > **Non-WM baselines.** Getting this direct comparison is actually surprisingly tricky, since WM papers report success rate while RL papers report reward (e.g. [[1](https://proceedings.neurips.cc/paper_files/paper/2024/file/6a5c38a730b17f3827c09cf6b192be04-Paper-Conference.pdf)], which also reports ~10m compute time for Push-T vs. <1m at our longest Push-T horizon). Regardless though, the comparison is a bit apples-to-oranges since we want to focus on zero-shot performance (measuring completion time from the first moment the WM is given a task), rather than evaluating a policy that has been trained for longer.
> > >
> > > That said, some close comparisons are LatCo (which we compare in our paper, note the WM’s they originally tested on have a much lower dimensional state space, 30-dimensional vs. >10,000 for DINO-WM) and the recent [adversarial WM](https://arxiv.org/abs/2512.09929) we mentioned to another reviewer, which reports at H=50 for Push-T that DINO-WM in a closed-loop setting achieves 16% success for the base DINO-WM and 26% for their adversarial WM, while GRASP achieves 43.4% on base DINO-WM but open-loop (their Table 9).
> > >
> > > **Single world model for all tasks:** There are recent works that train single world models on a greater collection of tasks simultaneously (e.g. the recent paper [Newt](https://arxiv.org/pdf/2511.19584)), but for this paper we follow the protocol of papers like DINO-WM and evaluate our planner on single-environment-trained world models that have expressive enough of a state feature space to be trained on many tasks (e.g. DINO features, which Newt also uses).
> > >
> > > We also hypothesize that GRASP’s advantage may grow as world models scale, since GRASP’s stop-gradient design is explicitly to avoid the adversarial robustness issues for collocation-based planners through deep learning-based world models, without sacrificing the prediction accuracy of the world model itself [[2](https://arxiv.org/pdf/1805.12152)]. We leave this empirical question for future work, and focus the current paper’s analysis on characterizing GRASP as a planner.
> > >
> > > If there are any remaining questions/concerns keeping you from raising your score, we are happy to address them in the time remaining. Thank you again for your careful review.

---

### Decision · Program_Chairs · 2026-04-30

**Decision:**

Accept (regular)

**Comment:**

GRASP presents parallelized gradient-based planning for vision-based world models by treating intermediate states as optimization variables with soft dynamics constraints, adapting classical direct collocation to learned models. All three reviewers give Weak Accept, recognizing clean formulation, strong long-horizon improvements over CEM/gradient descent baselines, and thorough ablations. The rebuttal provided adequate hyperparameter sensitivity analysis. Remaining concerns include limited evaluation on simple benchmarks (Push-T, PointMaze) without real-world or high-dimensional validation, and missing direct comparison to classical trajectory optimization methods beyond LatCo. The contribution is technically sound but incremental—adapting well-known optimization ideas to a neural setting.